

# Review article: The hydrology of debris-covered glaciers - state of the science and future research directions

Katie E. Miles[1*], Bryn Hubbard[1], Tristram D. L. Irvine-Fynn[1], Evan S. Miles[2], Duncan J. Quincey[2] and Ann V. Rowan[3]

1. Centre for Glaciology, Department of Geography and Earth Sciences, Aberystwyth University, Aberystwyth, UK

2. School of Geography, University of Leeds, Leeds, UK

3. Department of Geography, University of Sheffield, Sheffield, UK

* Correspondence to: kam64@aber.ac.uk

## Abstract

Debris-covered glaciers (DCGs) are characterised by distinct hydrological systems that differ fundamentally from those observed on clean-ice valley glaciers. To date, most studies of DCG hydrology have focused on supraglacial hydrology, given that surface streams are broadly accessible and repeat observations can lead to conceptual models of channel evolution. Few have characterised englacial conduits and their layout, and none have directly investigated potential subglacial drainage networks in any setting. In this review, we summarise the current state of knowledge relating to DCG hydrology with a global focus, and present our own field observations to illustrate the distinct nature of DCG landforms on a receding high-elevation glacier in the Himalaya. We draw on recent work that has gone some way towards providing a process-based understanding of the formation and evolution of englacial and subglacial hydrological pathways and consider the role that DCG hydrology plays in regulating water supplies to downstream communities, contrasting this information with clean-ice examples. We conclude by identifying important knowledge gaps that might be considered priorities for future research into DCG hydrology.

## 1. Introduction

Debris-covered glaciers (DCGs), also referred to as debris-mantled glaciers, moraine-covered glaciers or ice-cored rock glaciers, are present in nearly all of Earth's glacierised regions, with a particularly large concentration in the Himalayan mountain range (Bolch et al., 2012; Scherler et al., 2011). Ice and snow melt from these DCGs represents the source for some of the world's largest rivers: around 25% of Earth's population is dependent on glacier melt and seasonal snowpacks for drinking water, irrigation and hydroelectric power supplies (Immerzeel et al., 2010). The recent mass loss of glaciers in response to the warming climate is currently increasing river discharge and sea-level contributions (Lutz et al., 2014; Radić et al., 2014; Shea and Immerzeel, 2016), but studies simulating future scenarios are universal in predicting long-term reductions in flow, perhaps as



soon as 2050 in central Asia (Barnett et al., 2005; Bolch et al., 2012; Lutz et al., 2014; Ragettli et
al., 2016a; Sorg et al., 2012). This may threaten water security in many regions, particularly across
High Mountain Asia where most rivers source from glaciers in the Himalaya (Eriksson et al., 2009;
Hannah et al., 2005; Immerzeel et al., 2010; Winiger et al., 2005); these glaciers currently reduce
vulnerability to seasonal water shortages (Pritchard, 2017). A decreased discharge of the Indus and
Brahmaputra rivers alone is estimated to affect 260 million people (Immerzeel et al., 2010).
The long-term response of DCGs to changing climatic conditions is strongly non-linear and
reflects both spatial variability in debris concentration and climatic controls integrated over at
least several decades (Benn et al., 2012; Vaughan et al., 2013). Predictions of mass loss for
individual glacierised regions vary hugely. For example, in the Everest region of the Himalaya,
Rowan et al. (2015) predicted an 8-10% mass loss of glaciers by 2100, while Soncini et al. (2016)
calculated up to a 50% loss, and Shea et al. (2015) up to 99% loss in extreme scenarios (warming
of ~3°C). At a regional scale, model predictions also vary: Zhao et al. (2014) predicted a 22% total
loss of all glaciers in High Mountain Asia by 2050 (contributing 5 mm to sea-level rise); Chaturvedi
et al. (2014) found that up to 27% of glaciers in the Himalaya-Karakoram may have ablated
completely by 2080 under the most rapid warming scenario; and Kraaijenbrink et al. (2017) found
that 36% of glaciers in High Mountain Asia will be lost by 2100 with only a conservative 1.5°C global
temperature rise. Clearly, predictions such as these depend sensitively on the precise climate
scenario used, but a number of key knowledge gaps also exist concerning the character of DCGs
and the processes influencing them (Bolch et al., 2012; Huss, 2011). In particular, due to the
remoteness and inaccessibility of such glaciers, hydrological research has been severely limited.
In this review we consider the current state of knowledge of DCG hydrological systems, and
highlight key gaps as suggested topics for further research. While the review includes hydrological
research relating to DCGs located anywhere on Earth, it is noted that much of this research relates
to high-elevation Himalayan DCGs. First (Section 2), we discuss the formation and distribution of
DCGs. Next, we present a summary of existing research and understanding of the hydrological
systems of DCGs. This is considered in terms of four hydrological domains: supraglacial (Section 3),
englacial (Section 4), subglacial (Section 5), and proglacial (Section 6). Finally, in light of the above,
we propose several potential future research directions concerning the hydrology of DCGs (Section
66 7).

## 2. Debris-covered glaciers

Kirkbride (2011) defined a DCG as 'a glacier where part of the ablation zone has a
continuous cover of supraglacial debris across its full width'. While this definition has been broadly
adopted, we do not necessarily determine that the full width of the glacier terminus must be
debris-covered; however, debris must cover a large enough portion of this area to distinguish it
from a broad medial moraine (Anderson, 2000). Therefore, we define a DCG herein as 'a glacier
with a largely continuous layer of supraglacial debris that covers most of the ablation area, typically
increasing in thickness towards the terminus'. Figure 1 shows many of the common features of a
typical DCG, both from above (Figure 1A) and obliquely (Figure 1B).
DCGs are present in nearly all glacierised regions, and occur extensively where high rates
of rock uplift provide large amounts of sediment through glacial erosion in young mountain ranges



such as the Himalaya, Southern Alps and the Andes (Anderson and Anderson, 2016; Dunning et al.,
2015). Approximately 23% of all glaciers across the Himalaya-Karakoram have a debris cover
(Scherler et al., 2011), but due to the difficulties in mapping and accessing many glaciers, a global
map of DCGs has not yet been published (cf. Sasaki et al., 2016). DCGs have been mapped and
observed independently in the European Alps (Brock et al., 2010; Paul et al., 2004), Iceland (e.g.
Spedding 2000), Svalbard (e.g. Etzelmüller et al. 2001; Lukas et al. 2005), Scandinavia (e.g. Jansson
et al. 2000), Canada (e.g. Mattson 2000), Alaska (e.g. Kienholz et al. 2015), California (e.g. Clark et
al. 1994), the High Andes (Emmer et al., 2015; Janke et al., 2015; Racoviteanu et al., 2008), New
Zealand (e.g. Kirkbride & Warren 1999), Iran (e.g. Karimi et al. 2012), Caucasus (e.g. Stokes et al.
2007; Lambrecht et al. 2011) and Antarctica (e.g. Chinn & Dillon 1987; Levy et al. 2006; Mackay et
al. 2014). They are commonly found in areas of mountain permafrost (Schmid et al., 2015), while
permafrost-related patterned ground has also been observed on the debris layer of DCGs, for
example in Antarctica (Levy et al., 2006).
As well as on Earth, over 1,300 individual DCGs have been both identified (Baker, 2001;
Head and Marchant, 2003) and inventoried (Souness et al., 2012) in the mid-latitude regions of
Mars. Although currently colder and drier than Earth, Mars' so-called 'glacier-like forms' are
similarly lobate, debris-covered, deforming, and able to deposit debris to form bounding moraine
ridges. Both the detailed characterisation and broader dynamic glaciology of martian glacier-like
forms have been reported elsewhere (Hubbard et al., 2011, 2014), but given there are almost no
published data on their hydrological characteristics, planetary DCGs are not considered further
herein.
Debris is supplied to a glacier through avalanching, rockfalls and small landslides onto the
glacier surface (Figure 2A), thrusting from the bed, dust blown from exposed moraines or
solifluction from (ice-cored) moraines (Dunning et al., 2015; Evatt et al., 2015; Gibson et al., 2017a;
Hambrey et al., 2008; Kirkbride and Deline, 2013; Kirkbride and Warren, 1999; Rowan et al., 2015;
Spedding, 2000). Rockfall triggered by freeze-thaw processes (Nagai et al., 2013), landslides
(Hewitt et al., 2008) and permafrost degradation (Gruber and Haeberli, 2007) can also contribute
to the accumulation of debris on a glacier surface, and the frequency of such events appears to be
increasing with climate change (Gruber et al., 2004; Huggel et al., 2012). Where there is a supply
of debris in the accumulation zone, it is often advected into the ice and transported englacially
through the glacier along flowlines (Figure 2B); eventually being melted out at the surface in the
ablation area (Anderson and Anderson, 2016; Dunning et al., 2015; Evatt et al., 2015; Jansson et
al., 2000; Kirkbride and Deline, 2013). The surface of the accumulation zone is therefore commonly
largely free from debris, with a thin debris layer emerging at the surface near the equilibrium line
and increasing in thickness towards the terminus (Gibson et al., 2017b; Iwata et al., 2000). Debris
layers have been noted to develop more quickly, and to expand upglacier (or laterally from medial
moraines), during periods of glacier recession (Iwata et al., 2000). Debris can also be entrained
subglacially if it is frozen onto cold basal ice (Jansson et al., 2000) or where water is elevated under
pressure, triggering a switch from subglacial to englacial drainage and transporting debris up into
the glacier (Spedding, 2000).
A debris layer can range in thickness from a few millimetres, comprising scattered particles,
to several metres or more, comprising large rocks and boulders (Figure 2C & D) (Inoue and Yoshida,



1980; McCarthy et al., 2017). Direct measurements of thick supraglacial debris layers are difficult
to acquire, so published data are scarce. Gades et al. (2000) used radio-echo sounding on Khumbu
Glacier, Nepal Himalaya, to measure supraglacial debris up to 3 m thick; our own observations on
the same glacier suggest that in places the debris cover exceeds this thickness (Figure 2D). Satellite
imagery has also been used to approximate debris thickness in a variety of settings using debris
surface temperature measurements (Gibson et al., 2017b; Rounce et al., 2015): on Miage Glacier
in the Italian Alps, for example, the surface debris layer ranged from 0 to 0.6 m thick (Foster et al.,
2012; Mihalcea et al., 2008). Beneath the supraglacial debris layer, glacial ice can include entrained
debris (Figure 2B) or be debris-free (Figure 2C) (Schmid et al., 2015).
Several publications have reviewed the hydrology of 'clean-ice' glaciers (e.g. Fountain and
Walder, 1998; Hubbard and Nienow, 1997; Irvine-Fynn et al., 2011; Jansson et al., 2003) and ice
sheets (e.g. Greenwood et al., 2016). However, these reviews have omitted consideration of the
hydrology of DCGs, which is both under-investigated - and consequently very poorly understood -
and distinctive. This distinctiveness results from several characteristics, including: the presence of
supraglacial ponds that appear to interact intimately with near-surface englacial drainage; the
presence of a thick debris cover that influences patterns of surface melt and runoff; the possible
presence of cold ice advected from high elevation accumulation areas, influencing englacial and
subglacial drainage; the presence of a glacier tongue of low, or even reversed, surface slope that
would correspondingly influence the local hydraulic potential (Shreve, 1972); the common
presence of a substantial moraine-impounded proglacial lake that would also complicate near-
terminus englacial and subglacial water flows; and finally, the common location of DCGs in
monsoon-influenced areas, affecting temporal patterns of mass balance. Below, we summarise
the current information and understanding relating specifically to the hydrology of DCGs.

## 3. Supraglacial hydrology

Supraglacial hydrology includes meltwater generation, and meltwater transport through the
debris layer, supraglacial ponds, and supraglacial streams, eventually to be delivered to the
glacier's englacial drainage system or off the glacier margin directly to the proglacial zone. Here,
we discuss these flowpath components in sequence.

### 3.1 Meltwater generation

Similar to clean-ice glaciers, meltwater on DCGs is produced primarily through ablation of surface
ice and snow. However, the spatial pattern of the former is complicated by the presence of surface
debris over much of the ablation area of DCGs (Figure 2). Overall, the presence of thick debris
tends to suppress ablation. In the Caucasus, for example, debris layers were found to reduce melt
by an average of ~25% compared to clean ice (Lambrecht et al., 2011). This varies primarily with
the thickness and lithology of the debris layer (Figure 3). A debris layer thinner than a critical
thickness, typically of ~50 mm, will enhance albedo and thus increase the ablation rate compared
to debris-free ice. The ablation rate peaks at a debris thickness of ~2-5 mm, known as the
'effective' thickness  (Adhikary et al., 2000; Evatt et al., 2015; Inoue and Yoshida, 1980; Juen et al.,
2014; Lejeune et al., 2013; Nicholson and Benn, 2006, 2013; Østrem, 1959; Singh et al., 2000;
Takeuchi et al., 2000). A debris thickness greater than ~50 mm instead insulates the ice from

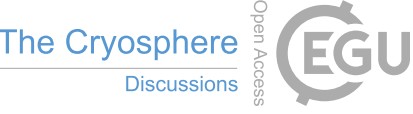

incoming solar radiation, increasing the albedo but inhibiting the penetration of excess surface
energy to the ice and thus reducing the melt rate (Figure 3). The exact values of the critical and
effective thickness are strongly dependent on the thermal conductivity of the debris (Figure 3),
which can vary widely across a glacier surface and differs according to whether the debris is wet
or dry (Casey et al., 2012; Collier et al., 2014, 2015; Gibson et al., 2017a; Nicholson and Benn, 2013;
Pelto, 2000). For example, Kayastha et al. (2000) found that maximum ice ablation occurs beneath
a debris layer that is 3 mm thick on Khumbu Glacier. Variations in ablation according to these
factors represent an important first-order control on glacier surface morphology, and are partially
responsible for the characteristic hummocky topography of large mounds separated by troughs
superimposed upon a concave surface profile of debris-covered surfaces (Figure 1B).
Beneath a debris depth of 250 - 300 mm, the ice becomes almost fully insulated from short-
term surface energy fluxes, and the storage and conduction of heat through the debris layer plays
a much greater role in the ablation that occurs (Bocchiola et al., 2015; Brock et al., 2010; Conway
and Rasmussen, 2000; Nicholson and Benn, 2013; Østrem, 1959; Reid and Brock, 2010). For
example, a thick debris layer comprised of fine-grained particles with a low void space reduces the
rate of evapotranspiration driven by air flow at the debris-ice interface, resulting in more energy
available for melt (Evatt et al., 2015). Conversely, the presence of moisture in debris layers >1 m
thick has been found to decrease the efficiency of heat transfer by decreasing the thermal
diffusivity of the debris layer, thus reducing heat transmission and melt of the glacier ice below
(Collier et al., 2014). Although melt rates beneath thick debris layers are low, they are thus non-
negligible and hence need to be considered in glacier-wide surface energy-balance calculations
(Collier et al., 2015).
The ablation rate of DCGs is enhanced by the presence of supraglacial ponds (Section 3.3)
and ice cliffs (Figure 2B & D) that are generally absent from equivalent clean-ice valley glaciers.
Supraglacial ice cliffs can form through the slumping of debris from steep slopes, calving at the end
of supraglacial ponds, or the collapse of englacial voids (Section 4); all of which expose steep, bare
ice faces at the glacier surface (Benn et al., 2001, 2012; Sakai et al., 2002; Thompson et al., 2016).
Ice cliffs contribute a notable proportion of the ablation of DCGs (Brun et al., 2016; Buri et al.,
2016a; Han et al., 2010; Juen et al., 2014; Reid and Brock, 2014; Sakai et al., 2000, 2002; Thompson
et al., 2016), accounting for up to 69% of the total ablation of debris-covered areas whilst covering
as little as 2% of the total glacier area, exhibiting melt rates often 10-14 times higher than beneath
debris-covered ice (Immerzeel et al., 2014; Sakai et al., 1998).
Where ice cliffs are associated with supraglacial ponds, there is further potential for
increased melt rates through undercutting and calving processes (Brun et al., 2016; Buri et al.,
2016b; Miles et al., 2016a; Röhl, 2008; Thompson et al., 2016). Combined ice cliff and pond systems
have been found to contribute significantly to the surface lowering of DCGs (King et al., 2017;
Nuimura et al., 2012; Pellicciotti et al., 2015; Ragettli et al., 2016b; Thompson et al., 2016; Watson
et al., 2017) and, since it tends to be larger cliffs (and hence greater potential areas for melt) that
are associated with ponds (Kraaijenbrink et al., 2016a), this could lead to more rapid glacier surface
lowering and meltwater production (King et al., 2017). A decadal trend of surface lowering,
stagnation and glacier mass loss has already been observed on a large number of Himalayan DCGs
(Bolch et al., 2011, 2012; Kääb et al., 2012; Pellicciotti et al., 2015; Scherler et al., 2011) as a result





of warmer air temperatures and weaker monsoons (Pieczonka et al., 2013; Thakuri et al., 2014).
Surface lowering rates measured at glaciers in the Everest region were as high as 1.62 ± 0.14 m a⁻
¹ for high-elevation land-terminating glaciers in the Pumqu catchment between 2000 and 2015
(King et al., 2017). Furthermore, surface lowering on DCGs is leading to an overall increase in debris
thickness (Gibson et al., 2017b) and an upglacier emergence of a thin supraglacial debris layer,
which will further increase albedo and surface meltwater production (and hence lowering,
potentially leading to a positive cycle until debris thickens sufficiently to insulate the surface)
(Kirkbride and Warren, 1999; Stokes et al., 2007), but may make observations of subsurface
hydrology even more difficult.
### 3.2 Debris layer hydrology
The occurrence of some ice ablation beneath even a thick debris layer implies that during much of
the ablation season, water must exist between the ice surface and the debris layer (McCarthy et
al., 2017), likely as a thin film. Subsequently, transport of this meltwater must occur, for example
as a saturated surface layer or – initially at least – as tiny rivulets. However, despite its importance
in contrasting with standard models of supraglacial hydrology based on research at clean-ice
glaciers, this process remains unexplored. This at least partly reflects the difficulty involved in
gaining non-influencing access to the ice-debris interface beneath thick surface debris. Despite the
absence of direct observations, meltwater transport through such a layer is likely to be slow and
inefficient, and water may be stored within the debris layer, introducing temporary delays in the
transport of meltwater through the system and thus affecting meltwater hydrochemistry (Tranter
et al., 1993, 2002), the development of other parts of the drainage network and the proglacial
discharge.
However, some parallels may be drawn from comparable systems, such as water flow
within debris above and the active layer of permafrost, moraines and talus fields, in order to
speculate how this transport may occur. For example, Hortonian overland (or infiltration excess)
flow is used to describe initial annual melt in permafrost regions, when frozen soils limit infiltration
producing a shallow saturated soil layer, above which overland flow is produced (Woo, 2012; Woo
and Xia, 1995). This has been observed within talus fields that are underlain by seasonally frozen,
and hence impermeable, ground (Liu et al., 2004) and within the active layer located beneath a
layer of debris, with meltwater infiltrating down to, then flowing downslope above the
impermeable permafrost table (Rist and Phillips, 2005). Where bedrock or debris is present, water
is transported through cracks or spaces between the rocks, but may also follow furrows between
linked depressions (Woo, 2012). A similar situation may hold on a smaller scale between
impermeable glacial ice and the overlying debris layer, with water flowing in runnels eroded either
down into the ice or between rocks in the debris layer. However, any such model remains to be
evaluated.
In general, water flow within and below the supraglacial debris layer and across the
impermeable supraglacial ice surface would be expected to be directed downglacier towards the
terminus and lateral margins (Winter, 2001). Clean-ice glaciers typically have a convex supraglacial
geometry, producing clearer watersheds and drainage routes. On DCGs, this pattern is complicated
by the presence of hummocky topography and a concave surface profile that commonly results
from the reversed mass balance gradient (Bolch et al., 2011), interrupting and complicating these





drainage routes (Benn et al., 2017; Miles et al., 2017). Although relatively unexplored, these factors
can lead to multiple scales of superimposed hydrological units, from a single supraglacial
depression to the full watershed hydrological unit (Winter, 2001).
Moraine-talus features in proglacial environments may also provide a shallow subsurface
flow system comparable to water within the supraglacial debris layer on DCGs. Investigation of a
moraine-talus feature containing buried ice at Opabin Glacier in the Canadian Rockies
demonstrated a system of small channels flowing over the buried ice within the moraine, through
the bedrock and talus field beyond the moraine (Roy and Hayashi, 2009). Langston et al. (2011)
reported that subsurface ice at the same glacier acted as an impermeable layer causing relatively
fast and shallow groundwater flow towards depressions within the proglacial moraine. The water
accumulated within these depressions, saturating sediments or surface water features and
enhancing the melt of the subsurface ice (Langston et al., 2011). Both of these situations could be
plausible within the debris layer of a DCG: meltwater contained within the layer could augment
the melt of glacier ice below, or it could initiate and contribute to supraglacial hydrological features
such as supraglacial ponds and streams.

### 3.3 Supraglacial ponds

Supraglacial ponds (Figure 4), a term here used to also encompass larger water bodies elsewhere
referred to as lakes, are extremely common and important features on DCGs, particularly those
with recent surface lowering. Ponds are generally absent from clean-ice valley glaciers, but are
prevalent on low-gradient areas of glaciers draining ice sheets; close to the margin the surface is
too steep for water to accumulate (Chu, 2014; Sundal et al., 2009). Similarly on a DCG, given a
water supply, the most important control on the location of supraglacial pond formation is the
slope of the glacier surface, with ponds being most prominent in areas with the lowest gradients
(Miles et al., 2016b; Quincey et al., 2007; Reynolds, 2000; Sakai, 2012; Sakai et al., 2000; Sakai and
Fujita, 2010; Salerno et al., 2012). A surface gradient of 2° or less promotes the development of
larger lakes; at slopes greater than this threshold, smaller isolated and transient ponds are more
likely (Miles et al., 2016b; Quincey et al., 2007; Reynolds, 2000). Salerno et al. (2012) additionally
found that the upglacier slope has an influence on pond formation, being inversely correlated to
the total area of lakes downglacier.
Glacier velocity and motion type exert less important controls over the location of
supraglacial ponds. An increase in lake concentration was reported towards the terminus of DCGs,
which is also characterised by low or very low surface velocities (Kraaijenbrink et al., 2016a; Miles
et al., 2016b; Quincey et al., 2007; Sakai, 2012; Salerno et al., 2012, 2015). A decrease in velocity
towards the glacier terminus, as well as ice inflow at flow unit confluences (Kraaijenbrink et al.
2016b), causes longitudinally compressive flow, which tends to close transverse crevasses and
englacial conduits and force water back to the surface, as well as limiting drainage from the glacier
surface (Kraaijenbrink et al., 2016a; Miles et al., 2016b). The thinning and stagnation of DCG
termini may additionally have resulted in enhanced melting beneath the debris layer, further
promoting the formation of ponds (Salerno et al., 2015; Thakuri et al., 2016).
Initial supraglacial pond growth occurs through subaqueous melting at the base of any
slight depression (Chikita et al., 1998; Mertes et al., 2016; Miles et al., 2016a; Stokes et al., 2007;



Thompson et al., 2012). Once water has accumulated and been warmed by incoming solar radiation, the pond becomes warmer than the surrounding ice. For example, Chikita et al. (1998) measured a maximum temperature of ~5°C at the surface of a supraglacial lake on Trakarding Glacier, Nepal Himalaya. Excess energy is thus available for further ablation both vertically and laterally where the pond water is in contact with ice, increasing the pond size, steepening marginal slopes and mobilising debris to expose bare ice (Stokes et al., 2007). Xin et al. (2012) observed on Koxkar Glacier, Tien Shan mountains, that meltwater at 0°C flowing into a pond initially cooled the surface layer, but gradually mixed with warmer, deeper layers and warmed to ~4°C. This increased the layer's density, causing it to sink and therefore move the warmer water towards the base of the pond, providing greater potential for additional subaqueous melting. In addition, wind-driven currents promote water circulation and vertical transfer of heat downwards, further enhancing basal melt of the pond (Chikita et al., 1998).

Many supraglacial ponds are surrounded by ice cliffs (Figure 4) where ponds can expand by subaerial melting and backwasting of the bare ice face (Röhl, 2008). Pond stratification and wind-driven currents may further enhance the subaqueous melt expansion of supraglacial ponds by triggering calving of the ice cliffs. The warm surface layer of the pond is disrupted by wind-driven currents, and where it come into contact with glacier ice, can undercut the cliff beneath the waterline. Progressive undercutting and thermo-erosional notch development may then lead to calving of the ice cliff face (Chikita et al., 1998; Kirkbride and Warren, 1997; Mihalcea et al., 2006; Miles et al., 2016a; Röhl, 2006, 2008; Sakai et al., 2009). Ice cliff calving occurs when the subaqueous melt rate exceeds the ice cliff melt rate; this is noted to be effective when the fetch is greater than 20 m and the water temperature is 2-4°C, though is possible at lower values (Sakai et al., 2009). Calving expansion is particularly effective at larger ponds (Röhl, 2008).

Calving events cause further mixing of pond layers, driving warmer surface water towards the base and again enhancing basal melting. Thompson et al. (2012) reported that the largest deepening rates of a supraglacial pond on Ngozumpa Glacier, Nepal Himalaya, occurred adjacent to the highest calving ice cliffs. Furthermore, when debris that has been heated by solar radiation falls into a pond, it contributes to the energy available for melt around the pond base (Thompson et al., 2012). Although shallowing of ponds can occur by sedimentation from inflowing water, this tends to be outstripped by growth caused by ablation (Thompson et al., 2012).

A pattern of supraglacial pond evolution has been observed on DCGs, primarily based on observations in the Himalaya. According to this model, supraglacial ponds form as 'perched ponds' that lie above the englacial drainage network (Benn et al., 2012). As these ponds increase in area and depth, they evolve from perched to base-level features, where the base-level is determined by the height at which water leaves the glacial system (usually the elevation of a spillway through the moraine at the glacier terminus or even the bed if water is transported there) (Mertes et al., 2016; Thompson et al., 2012). However, differing sub-catchments may have differing base-levels defined by other hydrological features such as moulins, which can result in a stepped hydrological cascade based on these local base-levels; alternatively, the presence of a groundwater system can produce a regional base-level. Where glaciers are in recession, an increasing number of supraglacial ponds will form and grow over time, creating a chain of terminus-base-level lakes that coalesce as each individually increases in area (Figure 1) (Sakai, 2012; Salerno et al., 2012). The

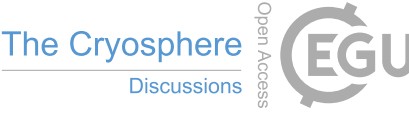



growth of base-level lakes is not limited by periodic drainage, and so such lakes can potentially increase exponentially in area, particularly through calving processes (Benn et al., 2001; Sakai, 2012; Thompson et al., 2012). If meltwater cannot escape from the system, lake expansion and coalescence will eventually lead to the formation of a single base-level moraine-dammed lake at the terminus (Mertes et al., 2016), that will then continue to expand both upglacier and downwards by ice melt. Water can escape the system by permeating through or flowing over the terminal moraine in a proglacial outlet spillway as the lake fills with sediment, or in rare instances the moraine dam may fail causing the lake to drain (Benn et al., 2001, 2012; Chikita et al., 2001; Sakai, 2012).

The progression of supraglacial pond evolution can currently be observed at various stages on many Himalayan glaciers. Several regions have experienced an increase in supraglacial pond area and proglacial lake formation in recent decades, assumed to be in response to a warmer climate and glacier surface lowering, for example glaciers in the Tian Shan mountains (Wang et al., 2013), Bhutan Himalaya (Ageta et al., 2000; Komori, 2008), Nepal Himalaya (Benn et al., 2000; Watson et al., 2016), New Zealand (Kirkbride and Warren, 1999; Röhl, 2008) and the Andes (Harrison et al., 2006; Rivera et al., 2007). Within the Hindu-Kush Himalaya, a clear divide has appeared between glacial lakes in the East, where there are a greater number of larger lakes that have grown progressively between 1990-2009 to become increasingly proglacial, compared to the western Himalaya, where smaller supraglacial lakes have generally been decreasing in area (Gardelle et al., 2011).

As isolated perched ponds widen and deepen, they can become connected to the englacial system by deepening to a point where they intersect englacial drainage channels and drain rapidly (Benn et al., 2001; Qiao et al., 2015; Röhl, 2008; Watson et al., 2016; Wessels et al., 2002), which temporarily halts the process of pond expansion (Mertes et al., 2016). Drainage then occurs periodically in a cycle of expansion and englacial connection, unlike larger, permanently hydraulically connected ponds, which tend to be more stable due to inputs of meltwater from streams and other ponds farther upglacier (Benn et al., 2001; Miles et al., 2017; Wessels et al., 2002). An abundant supply of meltwater from the ice surface or the wider drainage system is indicated by ponds with a high suspended sediment concentration (SSC); these ponds may also expand more rapidly due to the increased presence of warmer water for ablation of the pond walls (Takeuchi et al., 2012). Narama et al. (2017) observed a seasonal pattern of supraglacial pond filling and drainage, with 94% of their observed ponds over seven glaciers draining between 2013-2015. Pond seasonality has also been noted by Miles et al. (2016b), who found the maximum ponded area of five glaciers in the Langtang Valley, Nepal Himalaya, to occur during June for the study period 1999-2013. Larger ponds were also observed to partly drain and separate into multiple, smaller ponds, and later refill to form one large pond (Benn et al., 2001; Miles et al., 2016b; Wessels et al., 2002). Warmer temperatures during the spring months have been noted to correlate with a greater number of drainage events later in the same year, potentially due to the subsurface drainage system becoming increasingly connected from a greater amount of meltwater earlier in the year (Qiao et al., 2015).

Pond drainage is promoted in zones of higher local surface velocity and hence strain rates, creating a greater connectivity between the supraglacial and englacial drainage networks; more



frequent drainage in such regions results in smaller-sized ponds (Miles et al., 2016b). However, as
noted earlier in this section, ponds are more likely to form in areas with lower surface velocities.
Ponds may also drain by preferentially exploiting inherited structured weaknesses such as
sediment-filled crevasse traces, crevasses and englacial channels that have been forced closed in
regions of longitudinal compression, allowing drainage by hydrofracture (the penetration of a
water-filled crevasse through an ice mass assisted by the additional pressure of the water at the
crevasse tip)  (Benn et al., 2009, 2012, 2017; Gulley and Benn, 2007; Miles et al., 2016b).
Alternatively, perched ponds may drain by overspilling, when a channel is melted into the
downstream end of a pond; if, during drainage, this channel incises faster than the pond level
decreases then unstable and potentially catastrophic drainage can result (Qiao et al., 2015;
Raymond and Nolan, 2000). Analyses on Lirung Glacier, Nepal Himalaya, provided strong evidence
of continuous inefficient drainage of supraglacial ponds, likely into debris-choked englacial
conduits (Miles et al., 2017).

Supraglacial ponds are responsible for a large proportion of the melt from DCGs. Sakai et
al. (2000) estimated that ponds on Lirung Glacier absorb seven times more heat than the ice
beneath the debris-covered area, with at least 50% of this released with the melt output from the
pond. Miles et al. (2016a) found that subaqueous melt rates can keep pace with the backwasting
of ice cliffs, enabling these systems to propagate, and enabling the ice cliff to persist and backwaste
stably (Brun et al., 2016; Buri et al., 2016b). Both Sakai et al. (2000) and Miles et al. (2016a) inferred
that ponds have a strongly positive surface energy balance, and the warm water they discharge
contributes to internal melting along englacial conduits. This in turn leads, in some cases, to roof
collapse and the formation of new ponds (Benn et al., 2012; Miles et al., 2016a; Sakai et al., 2000),
resulting in a net glacier-wide increase in ablation rate. Salerno et al. (2012) stated that the
increasing presence of ponds is the clearest indicator of the effect that climate change is having
on DCGs.

## 3.4 Supraglacial streams

Supraglacial streams are commonly difficult to discern in debris-covered or crevassed regions of
the glacier surface, and therefore are rarely recorded in the literature. Large streams can be traced
in the upper reaches of some glaciers in satellite imagery, but small surface streams and diffuse
flows are less easily located and thus their prevalence remains unreported. For streams to form
and grow, a large catchment is required (Benn et al., 2017; Gulley et al., 2009a) and the rate of
stream downcutting must outpace the rate of surface lowering (Marston, 1983). Such conditions
may be promoted beneath thick debris with the ability to suppress surface ablation (Benn et al.,
2017). However, the presence of supraglacial streams has been recorded, ranging from small
temporary incisions to large perennial channels (Figure 5A-C). Stream growth and downcutting is
driven by thermal erosion (Marston, 1983) and can be marked by grooves down the side of the
channel showing previous high water-levels (Figure 5C); in extreme cases, ice cliffs form either side
of the stream. While such cliffs form on clean-ice glaciers, the relief of those on DCGs appears to
be more pronounced (Figure 5C & D), probably due to the debris-related suppression of surface
lowering away from surface streams.

Supraglacial streams have been noted originating in the upper ablation area of Khumbu
Glacier, for example beneath the ice fall, with at least one perpetual feature visible in several years





of satellite imagery (Gulley et al., 2009a). These streams seldom intersected with supraglacial ponds, instead progressively eroding into the debris layer with notable rates of downcutting: one stream was 5-10 m deep when it reached the lower ablation area where the debris layer is substantially thicker (Gulley et al., 2009a; Iwata et al., 1980). In this region, streams enter the glacier's interior; the nature of the entry point is unknown (Iwata et al., 1980). This is supported by our observations of a large supraglacial stream on Khumbu Glacier (Figure 5A & B) which begins to downcut into the glacier surface in the upper-mid ablation area and eventually disappears and becomes englacial in the mid-ablation area (Figure 5D), although the exact point of transition cannot be seen due to the presence of supraglacial debris and layers of older relict channels.

Similar to regions of clean-ice, supraglacial streams may drain into DCGs through crevasses or moulins (Gulley et al., 2009a; Iwata et al., 1980), or incise to a depth that they develop a closed roof through snow and debris accumulation combined with ice creep (Gulley et al., 2009a; Jarosch and Gudmundsson, 2012). However, supraglacial streams on DCGs differ from those on clean-ice glaciers due to the former's commonly reversed surface profile (Section 3.1). Such features are therefore often interrupted by crevasses or hummocky topography, and may not persist a long way along the glacier (Benn et al., 2017). Low surface gradients, low strain and longitudinal compression reduce the capacity for crevassing in the lower ablation area of heavily debris-covered tongues, where crevasses are therefore rarely encountered. Farther upglacier, often under conditions of strong longitudinal extension associated with ice falls, open crevasses are common and may suppress supraglacial stream development (Benn et al., 2017).

## 4. Englacial hydrology

There are relatively few observations of englacial drainage systems within DCGs, either directly or indirectly inferred. However, as most DCGs have a steep and variable surface gradient in their accumulation and upper ablation zones, crevassing can be prevalent, providing a route for supraglacial stream water to access the interior of the glaciers.

A glacier's thermal structure determines the water content of englacial ice, thereby exerting a primary control on the ability of an englacial hydrological system to form. Glacial ice can be defined as: cold (ice temperature below the pressure melting point); warm (temperature at the pressure melting point); or polythermal (zones of warm and cold ice). Glaciers with a polythermal structure can be further subdivided into several categories depending on the location of the boundary between the warm and cold ice (Blatter and Hutter, 1991). Glaciers with a higher warm ice content are more likely to contain a defined englacial hydrological system, as cold ice near the surface can limit, but not necessarily completely preclude, the penetration of meltwater into the glacial drainage system (Irvine-Fynn et al., 2011). Unfortunately, very few studies have determined the thermal structure of DCGs, and therefore little is known about whether and how water may route through these ice masses. Mae et al. (1975) measured an ice temperature of -5.3°C at 2.7 m depth within a borehole in the upper ablation area of Khumbu Glacier. By assuming that the ice temperature would increase with depth, they estimated the ice would reach pressure melting point at 16 m depth, and below this be warm-based to the bed (Mae, 1976; Mae et al., 1975). Similar assumptions were made for Rongbuk Glacier, Tibet, where ice temperatures were measured to a depth of 10 m (Academica Sinica, 1975). At a depth of 3 m, the ice temperature was



-4°C and continued to increase with depth. However, since none of these studies was able to measure temperature at a depth beyond that influenced by seasonal variations in air temperature (~10-15 m), the influence cannot be isolated. The assumption of a continued temperature increase to the pressure melting point with depth may also not be valid.

Techniques involving proglacial water properties have allowed some inferences to be drawn relating to the existence of englacial drainage systems within DCGs. Hydrological studies of surface mass balance components of Biafo Glacier, Karakoram Himalaya, allowed Hewitt et al. (1989) to infer water storage within the glacier at the start of the melt season, between the time of initial meltwater production and the subsequent reactivation/development of the drainage system. Hydrogeochemical analyses, particularly based on meltwater electrical conductivity (EC), have also been used to infer drainage pathways. Englacially-transported water has a lower sediment and ionic chemical content than subglacially-transported water, which entrains particles and solutes during contact with freshly-eroded basal debris, and therefore displays higher ionic concentrations, and hence EC values (Kumar et al., 2009). Consequently, studies have utilised this binary classification of low-EC supraglacially- and englacially-routed water as opposed to high-EC subglacially-routed water to attribute, via a mixing model, the proportions of water flowing through each system. For example, Hasnain & Thayyen (1994) used such a method to determine and differentiate between the englacial and subglacial components of the proglacial discharge of Dokriani Bamak Glacier, Garhwal Himalaya. Englacially, they found an efficient system that was active through the melt season, and that the amount of meltwater transport was proportional to supraglacial water production, implying a direct link between these systems. However, despite the evident utility of this approach, the assumption underpinning such mixing models has been questioned. Glacier drainage systems are inherently more complex than comprising only two principal pathways, and the solute content and degree of subglacial weathering can vary at a number of timescales (Sharp et al., 1995).

Englacial water storage and transport has been inferred from measurements of supraglacial pond water-levels, and an assumed connection between the pond and an englacial channel. Thakuri et al. (2016) measured a constant water-level in Imja Tsho lake on Imja Glacier, Nepal Himalaya, after the melt season, despite reduced precipitation and air temperatures, implying decreasing meltwater production. The authors attributed this to a lake recharge from englacially- and subglacially-stored water that was being progressively released over time. Further, the repeated filling and drainage cycle of perched ponds suggests that englacial conduits may play an important role in perched lake life cycles (Benn et al., 2017; Miles et al., 2017). Narama et al. (2017) found that the seasonal drainage cycle of supraglacial ponds on seven glaciers in the Tien Shan was characterised by a connection to an established englacial drainage system later in the summer: 94% of lakes drained and connected to an englacial system on all three years studied. Benn et al. (2012) proposed that the influx of large volumes of monsoon precipitation during the summer months may result in the opening of englacial (and subglacial) conduits, leading to the potential for considerable englacial ablation, subsequently calculated by Miles et al. (2016a) to be ~2600 $m^3$ for a surface pond of 500 $m^2$ over a single monsoon season.

There have been a number of direct observations of englacial channels using glaciospeleological techniques to explore and map conduits, and subsequently formulate theories





of channel development. Glaciolospeleology has been carried out on several DCGs primarily in the
Nepal Himalaya, including Khumbu Glacier (Gulley et al., 2009a), Ngozumpa Glacier (Benn et al.,
2009, 2017; Gulley and Benn, 2007), Ama Dablam and Lhotse Glaciers (Gulley and Benn, 2007),
and several DCGs in the Tien Shan (Narama et al., 2017). These investigations have provided direct
confirmation of a linked supraglacial-englacial system, often created by the drainage of
supraglacial ponds through englacial conduits (Gulley and Benn, 2007; Narama et al., 2017). On
Southern Inylchek Glacier, Tien Shan mountains, Narama et al. (2017) discovered both short
englacial channels linking chains of supraglacial ponds and longer channels with steeper gradients
extending from surface moulins. The latter may occur at the hydrological base-level of the glacier,
and show multiple levels of incision from progressive supraglacial pond drainages over time as the
base level has been eroded downwards (Gulley and Benn, 2007). Similar observations have been
made on Khumbu Glacier (Gulley et al., 2009a); however, Gulley & Benn (2007) noted that conduits
at different elevations may have varying local base-levels (Section 3.3). In such scenarios, this may
suggest that a subglacial drainage system either does not exist beneath the glacier, or is not linked
to the englacial system, or re-emerges to base level. Such scenarios, however, remain unreported.

Three formation mechanisms for englacial channels within DCGs have been proposed,
primarily from glaciospeleological investigations (Gulley et al., 2009a):
(I)    'Cut-and-closure' type conduits begin as supraglacial streams that incise downwards
over time, followed by roof closure through ice creep and supplemented by filling with
snow, ice and debris (Jarosch and Gudmundsson, 2012). This process requires a high
meltwater discharge such that downward incision is more rapid than glacier surface
ablation. Under such conditions, downcutting will continue to the hydrologic base-level
of the glacier (Gulley et al., 2009a). Cut-and-closure type conduits have been reported
by Gulley et al. (2009) on Khumbu Glacier, and Thompson et al. (2012) on Ngozumpa
Glacier. These conduits may be subject to repeated cycles of abandonment and
reactivation as water supply varies through the year, with abandoned channels closing
by ice creep. However, such channels rarely close completely due to their shallow
depth, and may be filled with sediment traces which provide lines of secondary
permeability by which the channel can be reactivated (Benn et al., 2009; Gulley and
Benn, 2007; Gulley et al., 2009a).
(II)   Meltwater may aggregate to form englacial channels by exploiting lines or planes of
secondary permeability; for example those left by relict cut-and-closure channels, or
debris-filled and/or compressed former surface crevasses (Benn et al., 2012; Gulley and
Benn, 2007; Gulley et al., 2009b). This may also be one mechanism by which perched
supraglacial ponds can drain (Miles et al., 2017). Along these low-permeability zones,
discharge through the icy matrix leads to the development of enlarging lines of
preferential flow due to viscous dissipation, eventually forming a phreatic conduit
(Benn et al., 2012).
(III)  Englacial channels may also form by hydrofracturing (Benn et al., 2009, 2012; Gulley et
al., 2009b). However, this is considered to be uncommon on DCGs as it requires surface
runoff to enter an open crevasse and is therefore generally restricted to elevations
above the debris-covered areas of DCGs (Benn et al., 2012). Nonetheless, channel
formation by hydrofracturing has been reported on Khumbu Glacier, where the



channels formed within a region of strong transverse extension that resulted in the
formation of longitudinal crevasses (Benn et al., 2009, 2012). In such zones, repeated
hydrofracturing is encouraged by the combined effect of elevated water pressure in
the base of a supraglacial lake with transverse stresses, producing  successively lower
niches in the walls indicating multiple stages of hydrofracturing followed by channel
closure by freeze-on (Benn et al., 2009).
Longer-distance water transport has also been observed through perennial sub-marginal
channels located along the edge of DCGs, likely formed by cut-and-closure of supraglacial channels
(Benn et al., 2017; Thompson et al., 2016). Gulley & Benn (2007) suggested that such marginal
features could provide longer-distance and more hydraulically efficient pathways than shallower
englacial conduits that occur more centrally within the glacier, due to the frequent presence of
infilled crevasse traces that can be exploited by water flowing at the margins. Centrally-located
englacial conduits are more likely to be discontinuous in nature, as a result of enhanced surface
lowering which can expose part of a conduit and re-route the water back to the surface (Figure 6)
(Miles et al., 2017).
Englacial conduits within DCGs may increase in efficiency through the melt season, as water
transported through the channels provides additional energy for melt and consequently erodes
the channel walls (Miles et al., 2016b, 2017; Sakai et al., 2000). For englacial channels located near
the surface, rapid expansion can result in conduit collapse as the roof is not sufficiently supported,
with the conduit walls forming ice cliffs and contributing to more rapid surface lowering of the
glacier surface (Benn et al., 2017; Kraaijenbrink et al., 2016a; Miles et al., 2016a; Sakai et al., 2000;
Thompson et al., 2016, 2012). Pond drainage events can further accelerate this process, as well as
adding to the total glacier mass loss as the drained water conveys large amounts of energy,
contributing to more rapid erosion of the conduit walls (Sakai et al. 2000; Benn et al. 2012; Miles
et al. 2016a; Thompson et al. 2016). Rounce *et al.* (2017) observed an outburst flood at Lhotse
Glacier, Nepal Himalaya, which they attributed to be at least partly triggered by the release water
stored in englacial conduits that became overburdened during the transitional pre-monsoon
season, when meltwater production is increasing and the subsurface hydrology is not fully
developed.
Englacial conduit collapse, or closure in areas of transverse compression, can provide new
depressions for supraglacial ponds to form, or facilitate the formation of larger lakes (Benn et al.,
2001, 2012; Kirkbride, 1993; Kraaijenbrink et al., 2016a; Sakai et al., 2000; Thompson et al., 2012).
Conduit collapse results in new bare ice faces, including ice cliffs, where melt rates will be
enhanced and the depression may become flooded by that increased meltwater production
supplementing inputs from upglacier (Benn et al., 2012). The enhanced ablation of both the
meltwater retained within the depression, and the surrounding newly-formed ice cliffs (the old
channel walls), will accelerate the melt and surface subsidence of the glacier (Thompson et al.,
2012).

## 5. Subglacial hydrology

Almost nothing is known about the subglacial drainage of DCGs due to the difficulties in accessing
such systems, resulting in no direct measurements to date. Further, the existence of base-level



englacial streams and a perched water table are highly likely to complicate the detection of, and
distinction between, englacial and subglacial systems, at least approaching the terminus of DCGs.
Furthermore, the majority of reported DCGs terminate in ponds as a result of progressive surface
lowering (Section 3.3). This both increases the likelihood of some form of subglacial drainage but
at the same time reduces the likelihood of that system being channelised and severely hampers
its direct access. An additional complication arises from the high-elevation of some DCG source
areas, making it possible that the ice may be too cold throughout for water to penetrate to the
bed.
Nonetheless, remote sensing-based (Quincey et al., 2009) and field-based GPS
(Bartholomaus et al., 2008, 2011) studies of DCG surface velocities have inferred the occurrence
of basal sliding, which requires the presence of lubricating meltwater at the ice-bed interface.
While cold-based glaciers are frozen to the bed and move primarily by internal ice deformation
and creep (Glen, 1955; Nye, 1957; Weertman, 1983), glaciers with warm basal ice conversely have
water present at the bed, partly from pressure melting, and can move at greater speeds through
an additional basal motion component (Kamb, 1970; Nye, 1969; Weertman, 1957), either at the
rock-ice interface or from deformation of soft sediment (Boulton and Hindmarsh, 1987; Walder
and Fowler, 1994). Relatively rapid surface velocities, most notably in the central areas of glaciers
and in the summer months (when melting and rainfall delivery are greatest) (Figure 7) have been
recorded, for example, by Copland et al. (2009) who recorded a maximum velocity of >200 m a$^{-1}$
on the South Skamri Glacier, Pakistan Karakoram. Such velocity increases have been interpreted
as indicative of basal motion lubricated by the presence of subglacial meltwater (Copland et al.,
2009; Kääb, 2005; Kodama and Mae, 1976; Kraaijenbrink et al., 2016b; Kumar and Dobhal, 1997;
Mayer et al., 2006; Quincey et al., 2009). For some high-elevation DCGs, whose ablation areas
often include the base of ice falls, heavy crevassing may provide one route for water to access the
internal and basal drainage system of the glacier even if the ice is too cold for an englacial system
to reach the bed (Kodama and Mae, 1976). Although rare, such pathways can persist and lead to
the formation of moulins, which have occasionally been observed in the upper ablation areas of
glaciers, for example on Baltoro Glacier in the Pakistan Karakoram (Quincey et al., 2009), and
provide a direct connection to the bed. If meltwater can penetrate to the bed, it not only suggests
that the basal conditions are above the pressure melting point (at least locally), but that subsurface
hydrological systems are possible and even likely. However, to date very few measurements of the
internal or basal ice temperature of DCGs have been made (Section 4).
Further support for the existence of channelised subglacial drainage, at least near the
terminus of DCGs, is provided by the presence of single outlet channels at such glaciers. These also
discharge large volumes of heavily debris-laden water implying that the water had been
transported along the bed, entraining sediment (Quincey et al., 2009). On Ngozumpa Glacier, Benn
et al. (2017) interpreted spatially localised seasonal variations in glacier surface velocity as basal
sliding and inferred from this the presence of channelised subglacial drainage in the lower 10 km
of the glacier. Whether these fluctuations resulted from basal sliding and/or subglacial till
deformation is unknown in the absence of knowledge of subglacial conditions at the glacier (Benn
et al., 2017).





The existence of active subglacial drainage has additionally been inferred from bulk
meltwater analysis. Hasnain & Thayyen (1994) used EC measurements of the proglacial discharge
of Dokriani Bamak Glacier to argue for a perennially-active subglacial system that is interconnected
with the englacial system. On the same glacier, Hasnain et al. (2001) used dye-tracing studies to
investigate the subglacial drainage system, inferring a possible switch between an inefficient and
an efficient drainage system, as has been observed on lower-elevation alpine glaciers (Mair et al.,
2002; Nienow et al., 1998). The same methods were used on Gangotri Glacier, Garhwal Himalaya,
to show that an efficient channelised system exists at atmospheric pressure and develops through
the melt season with increasing meltwater inputs (Pottakkal et al., 2014). Wilson et al. (2016)
inferred a large amount of subglacial meltwater storage on Lirung Glacier, compared to the debris-
free Khimsung Glacier, Nepal Himalaya, due to the smaller magnitude of diurnal discharge
variability from the former. Bhatt et al. (2007) measured higher solute ($Ca^{2+}$ and $SO_4^{2-}$)
concentrations in the proglacial discharge of Lirung Glacier compared to those in the supraglacial
ponds on the glacier, inferring that these chemical species were acquired through contact with
reactive debris during subglacial drainage.
The particle-size distribution of sediment suspended within the proglacial stream of
Gangotri Glacier was interpreted in terms of the waterborne evacuation of subglacially-eroded
fines (Haritashya et al., 2010). Both the net flux and size of the suspended particles increased
through the melt season, implying that the glacier's drainage system became progressively more
competent and interconnected through the melt season.
Thus, although several lines of investigation point to the likely existence of subglacial
drainage beneath DCGs, evidence to date has been invaluable, but – in the absence of first-hand
access – necessarily inferential and ambiguous.

## 6. Proglacial hydrology

### 6.1 Proglacial lakes and GLOFs

Proglacial lakes predominantly form by a continuation of the processes of glacier thinning and
supraglacial pond growth, as described in Section 3.3. Perched supraglacial ponds grow both
downwards, eventually cutting to base-level, and laterally, with many lakes eventually coalescing
to produce one large lake above and over the terminus (Figure 8) (Kattelmann, 2003; Mertes et
al., 2016; Röhl, 2008; Watanabe et al., 2009). Base-level lakes that penetrate the full glacier
thickness can form farther upglacier and expand downglacier through the isolated stagnant
terminus ice, for example Imja Lake on Imja Glacier (Watanabe et al., 2009), though this is less
common and perhaps reflects stagnant ice towards the terminus acting as a flow impediment. The
exact location of such a proglacial lake may also be determined by the location of shallow englacial
conduits that provide pre-existing lines of weakness as the perched ponds grow (Benn et al., 2017;
Thompson et al., 2012). Proglacial lakes will therefore be at the hydrological base-level of the
glacier, and are often dammed by the terminal moraine (Thompson et al., 2012). With time, such
lakes continue to erode downwards into the ice, eventually reaching bedrock or basal sediment.
Hooker Lake, New Zealand Southern Alps (Figure 8), which initially formed in 1994 in front of
Hooker Glacier is now approximately 2.5 km long, 500 m wide, and has a maximum water depth
of 140 m (Robertson et al., 2012).

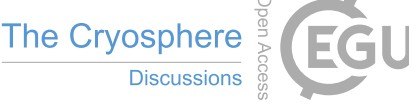

The formation of moraine-dammed proglacial lakes characterises a further and final stage
in the surface lowering and overall mass loss of DCGs. Benn et al. (2012) defined three stages in
the development of DCGs: in regime one, all parts of the glacier are dynamically active; in regime
two, surface lowering has begun and ice velocities decrease; in regime three, glaciers are
completely stagnant and rapid recession may occur. The formation of base-level lakes indicates
that a glacier has entered this third regime, and rapid recession may then occur through further
expansion of this proglacial lake (Benn et al., 2012). An increasing number of lakes of increasing
size have been observed in recent decades around the world (Carrivick and Tweed, 2013), for
example in the Caucasus Mountains (Stokes et al., 2007) and across the Himalaya (Gardelle et al.,
2011; Thompson et al., 2012). However, the pattern of proglacial lake formation has varied across
the Himalaya, with glacial lake coverage in the western Himalaya decreasing 30-50% from 1990-
2009 compared to an increased area of 20-65% in the eastern Himalaya, concurrent with the much
greater observed glacial mass loss in the former region over this period (Gardelle et al., 2011).
Proglacial lakes continue to expand through similar mechanisms to supraglacial ponds until
they are limited by subglacial topography, enhancing glacial mass loss and thus meltwater
production where the lake is underlain by ice (Carrivick and Tweed, 2013; Röhl, 2008). Initial
growth occurs through subaqueous melting and subaerial ice-face melting, causing both
deepening and areal growth, but once calving is triggered it becomes the dominant method of lake
growth (Röhl, 2008; Thompson et al., 2012). Calving from a proglacial lake progresses from notch-
development and roof collapse to large-scale, full-height slab calving that can substantially
increase mass loss from a glacier (Kirkbride and Warren, 1997; Thompson et al., 2012). If the lake
deepens to the glacier bed, allowing full-height slab calving, the lake may become unstable
because the water depth will be sufficient to trigger extending flow in the now-unsupported ice
cliff (Kirkbride and Warren, 1999; Thompson et al., 2012). This may weaken the ice by forming
crevasses, and allow the ice cliff to calve at a faster rate again; several kilometres of such rapid
calving was reported by Kirkbride & Warren (1999) for Tasman Glacier, New Zealand Southern
Alps. The process could also result in an upglacier expansion of the lake (Watanabe et al., 2009),
which may have implications for the glacier's drainage system, such as by earlier interruption of
meltwater routing (Carrivick and Tweed, 2013).
Very large proglacial lakes can alter a glacier's microclimate, due to a lake's lower albedo
and higher thermal heat capacity relative to surrounding ice and soil surfaces, producing relatively
cooler summer air temperatures and warmer autumn temperatures (Carrivick and Tweed, 2013).
This can slow summer ice ablation and consequently reduce the amount of meltwater being
produced and transported through the glacier, with implications for the development of englacial
and subglacial drainage systems. If a moraine-dammed proglacial lake is present then the
overwhelming majority of water transported through a DCG will pass through it (Benn et al., 2017).
This has implications for water drainage through the glacier, and for the potential occurrence of
glacial lake outburst floods (GLOFs) if the lake overflows or the dam is breached.
GLOFs can be a major hazard in regions such as the Andes and Himalaya, and can result in
fatalities as well as the destruction of land and infrastructure (Richardson and Reynolds, 2000;
Rounce et al., 2016). GLOFs can either occur through a breach of the dam or by dam failure. Dam
breach can be triggered by the increase of lake water-level and/or the creation of waves through:



the addition of water from a lake higher up on the glacier (Buchroithner et al., 1982); an ice
avalanche (Vuichard and Zimmermann, 1987); a rock avalanche or mass movement entering the
lake (Harrison et al., 2006; Rounce et al., 2017); glacier calving (Kattelmann, 2003); rainfall events,
particularly during the monsoon (Kattelmann, 2003; Osti et al., 2011); or an earthquake-triggered
overtopping (Rounce et al. 2016).
The second mechanism by which a GLOF can occur is through dam failure, of either an ice-
cored or a sediment-cored moraine. Ice-cored moraine dams are inferred to be common features
at DCG proglacial lakes, as dead ice can be left beyond the glacier terminus as a result of both
glacier retreat and differential mass wasting (Richardson and Reynolds, 2000). Ice-cored moraines
degrade progressively by ablation beneath the debris layer and from the warmer lake water; this
accelerates once the ice is exposed and subjected to enhanced aerial melt, and the dam may finally
fail when water routes through relict glacial drainage features, such as voids, reducing the dam's
structural strength (Kattelmann, 2003; Richardson and Reynolds, 2000). Non-ice-cored moraines
are entirely composed of glacial sediment, and have been observed to destabilise and fail as a
landslide after rainfall or earthquake events (Osti *et al.* 2011). Waves generated by the dam breach
mechanism can also initiate rapid erosion of either type of moraine (Hubbard et al., 2005), possibly
eventually triggering moraine failure (Kattelmann, 2003).
The onset of DCG recession by rapid calving could allow major rock and debris avalanches
into a proglacial lake, which could trigger a GLOF and potentially destabilise a mountainside, with
the possibility of further hazards such as landslides and rockfalls (Hubbard et al., 2005; Kirkbride
and Warren, 1999). Risks from GLOFs can be mitigated, for example, by artificially lowering the
proglacial lake water-level (Rana et al., 2000), or monitored using on-site or remotely sensed data
(Bajracharya and Mool, 2009; Bolch et al., 2008; Nie et al., 2013; Rounce et al., 2016; Watson et
al., 2015). As an increasing number of receding glaciers form a proglacial lake that not only
withholds proglacial discharge, but dams it up against a potentially unstable moraine-dam, the
possibility of devastating GLOFs could rise (Carrivick and Tweed, 2013; Gardelle et al., 2011; Stokes
et al., 2007; Thompson et al., 2012).

## 6.2 Proglacial streams

Proglacial runoff from DCGs can form a significant proportion of the discharge of large rivers
downstream, particularly in High Mountain Asia: the Indus, Dudh Koshi, Ganges and Brahmaputra
rivers all stem from glacial meltwaters (Pritchard, 2017; Ragettli et al., 2015; Wilson et al., 2016).
Proglacial discharge measurements, estimates and models have been made across the Himalaya,
for example on individual glaciers in Nepal (Braun et al., 1993; Fujita and Sakai, 2014; Ragettli et
al., 2015; Rana et al., 1997; Savéan et al., 2015; Soncini et al., 2016; Tangborn and Rana, 2000),
Tibet (Kehrwald et al., 2008), the Tien Shan (Caiping and Yongjian, 2009; Han et al., 2010; Sorg et
al., 2012), India (Hasnain, 1996, 1999; Khan et al., 2017; Singh et al., 1995, 2005; Singh and
Bengtsson, 2004; Thayyen and Gergan, 2010), and for multiple catchments and entire regions
(Winiger et al., 2005). However, few of these measurements have been made for longer than a
decade: of the studies listed above, five measure discharge for a year or less; three have 2-3 years
of measurements; and only one has 6 years of measurements; the rest use modelling to obtain
estimates of proglacial discharge. Although Pritchard (2017) found that the glacial contribution to

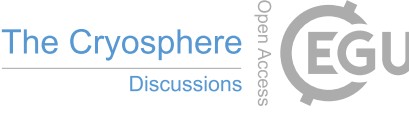



seven river basins in the Himalaya is proportionally small (0.1-3.0%) it increases upstream and was argued to be vital to support the freshwater needs of millions of people.

The presence of surface debris can have a notable effect on the proglacial discharge of a DCG, resulting in a proglacial hydrograph that is different from that of a clean-ice glacier (Figure 9). For example, discharge both diurnally and through the ablation season are muted at debris-covered Dome Glacier, Canadian Rockies, compared to neighbouring clean-ice Athabasca Glacier (Figure 9), producing an annual variance in volumetric discharge of 1% compared to 24% respectively (Mattson, 2000). This is due in part to the suppression of surface melt by a debris cover, and in part to the lags that are induced as a result of the debris layer. On a clean-ice glacier, the maximum melt rate occurs close to the time of maximum incoming solar radiation. Conversely, on a DCG, the additional time to conduct heat through a debris layer and the warmer local air temperatures due to the warming debris introduces a delay. Thus, peak melt can occur up to several hours after the maximum radiation receipt at the debris surface (Carenzo et al., 2016; Conway and Rasmussen, 2000; Evatt et al., 2015), and has been measured occurring up to 24 hours later for debris layers >0.85 m thick (Fyffe et al., 2014). This lag in diurnal peak melt is thus reflected in the timing of peak stream flow, producing a later and less pronounced peak in a proglacial stream's diurnal pattern (Fyffe et al., 2014).

Lags in the proglacial discharge at DCGs are also caused by the temporary storage of water within the debris layer. This has been observed during rainfall events and has been suggested to influence discharge through both the subglacial and proglacial drainage networks by delaying and buffering water transfer at the surface, potentially affecting basal water pressures and minimising peaks in proglacial discharge (Brock et al., 2010). However, in the Himalaya, the monsoon precipitation is thought to exert only a weak control on the proglacial discharge hydrograph of glaciers unless the intensity is >~20 mm d$^{-1}$, which occurred on 20% of rainfall days during four years of monsoon measurements (Thayyen et al., 2005). Early in the melt season, meltwater is additionally stored within the snowpack of DCGs as well as within the debris layer year-round, providing a further delay in the transport of meltwater from the surface into the subsurface drainage system (Singh et al., 2006b). However, in the last two decades the amount of snowfall accumulation has decreased across the Himalaya, and is projected to decrease a further 20-40% by 2100 (Salerno et al., 2015; Viste and Sorteberg, 2015) which is highly likely to reduce this buffer and influence the proglacial hydrograph pattern of DCGs in the future.

Groundwater storage within glacial catchments has been inferred to interact with proglacial (and subglacial) stream networks, affecting the discharge patterns of the streams due to additional water storage and subsequent release (Gremaud et al., 2009; Smart, 1988, 1996). Andermann *et al.* (2012a) observed a lag between precipitation and discharge for 12 Himalayan catchments (both glacierised and non-glacierised), indicating that up to two-thirds of the river discharge is stored for approximately 45 days in a groundwater aquifer system before the monsoon, greatly affecting the annual discharge pattern. This has been recorded in further studies measuring SSC, with much lower concentrations measured post-monsoon once this groundwater begins to be released and reduces the SSC of these rivers (Andermann *et al.*, 2012b; Andermann *et al.*, 2012c). Such a significant effect of groundwater storage and release downriver from DCG catchments would suggest that similar processes may occur beneath the glaciers themselves.



Other studies of glacierised limestone karst aquifers have used dye-tracing and modelling
to investigate links to the glacial drainage system. At Glacier de la Plaine Morte, Swiss Alps, this
showed that a greater proportion of the glacial meltwater was transported through a karst system
during the winter; in the summer, the karst capacity was exceeded and the excess water drained
through the glacier instead (Finger et al., 2013). A similar system in the Jade Dragon Snow
Mountain region of southwest China was studied for stable isotopes and modelled by Zeng *et al.*
(2015), showing that 29% of the glacier meltwater was transported into the karst aquifer.
Groundwater sinks of subglacial meltwater can therefore comprise a significant portion of the total
glacial output, potentially resulting in the glacial ablation being underestimated if this is not taken
into account.

As DCGs provide a significant source of water for large populations, quantifying future
runoff volumes is vital for planning and mitigating water resource issues. Models have been used
to predict future runoff from DCGs for a single glacier basin (Ragettli et al., 2015; Singh et al.,
2006a, 2008; Zhang et al., 2007), and multiple glacier basins (Immerzeel et al., 2012; Lowe and
Collins, 2001) up to a regional scale (Rees and Collins, 2006; Shea and Immerzeel, 2016),
investigating various future climatic scenarios. Currently, a large proportion of DCGs worldwide,
particularly in the Himalaya, have negative mass balances (Bolch et al., 2011, 2012; Kääb et al.,
2012; Scherler et al., 2011). The projected decrease in snowfall will additionally contribute to the
decreasing mass of these glaciers, both by reducing accumulation rates but also by exposing the
glacier surface to atmospheric melting earlier in the melt season (Salerno et al., 2015). Glacier
contributions to catchment discharge in many regions have been predicted to increase over the
next few decades, but as the glaciers continue to shrink, this proportion will begin to reduce
substantially due to the significantly smaller volume of glaciers remaining (Barnett et al., 2005;
Bolch, 2017; Bolch et al., 2012; Huss, 2011; Lutz et al., 2014). Shea & Immerzeel (2016) estimated
that most basins will have declining glacier contributions to streamflow by 2100, and water
shortage may then be a concern for many populated areas in the Karakoram, while peak flows may
represent a greater concern in the eastern Himalaya.

A further concern for future water supplies is the water quality provided by glacial
discharge, which is commonly assessed through measurements of the EC and SSC of proglacial
streams. Although based on simplified mixing models, studies have used proglacial stream SSC to
calculate the contribution of glacial systems to overall catchment sediment yields (Collins 1996;
Collins 1999; Hasnain & Thayyen 1999a; Singh et al. 2005; Haritashya et al. 2010). For example,
Collins (1996) determined from investigations at Batura Glacier, Karakoram Himalaya, that 40% of
the sediment yield of the Indus river, and 60% of the Hunza river are glacially-derived. Tectonic
uplift also contributes through enhanced weathering to the high sediment flux in these regions
(Collins, 1996). However, the glacially-derived proportion of total sediment yield can vary widely
with, in general, glaciers with more extensive subglacial systems and higher discharges
contributing greater amounts of sediment (Collins, 1999). Proglacial SSC therefore increases with
discharge during the ablation season, particularly with monsoon rainfall (Collins 1999; Hasnain &
Thayyen 1999a) when supraglacial debris weathering is enhanced and the increased discharge
flushes sediment through the system, increasing chemical weathering rates (Hasnain & Thayyen
1999b; Hodson et al. 2002) which may have implications on the water quality downstream as
discharge increases with glacier mass loss. Although the monsoon rains contribute to enhanced





sediment transport, they are not considered to affect weathering within the subglacial systems of
such glaciers, where sulphide oxidation and calcium carbonate dissolution dominate (Tranter et
al., 2002). On a diurnal scale, Kumar et al. (2009) found that the total ion concentration of
proglacial meltwater increased from the afternoon onwards, as the (inferred) englacial and
subglacial systems of Gangotri Glacier, became more active.
The water quality of proglacial runoff, including carbon export and other nutrient delivery
from glacial basins, exerts a critical influence on biogeochemical fluxes, ecosystem services,
downstream ecology and aquatic ecosystem biodiversity (Jacobsen et al., 2012). Ecological
responses are extremely sensitive to reductions in glacier area, with studies finding that freshwater
biodiversity in glacier-fed streams will decrease rapidly with the reduction (and ultimate
disappearance) of glacier area (Cauvy-Fraunié et al., 2016; Jacobsen et al., 2012; Milner et al.,
2009; Wilhelm et al., 2013). The potential loss of species is a key issue for future conservation and
the evolution of glaciers, particularly DCG, will have a large influence on any loss of species
(Jacobsen et al., 2012), an area of study largely beyond the remit of this review, but which deserves
further investigation.

## 843    7. Summary and future research priorities

The hydrology of DCGs is sufficiently distinctive to warrant bespoke treatment, separate from that
of clean-ice valley glaciers. This distinctiveness stems principally from the extensive and thick
debris cover on DCGs as well as, in many cases, their high elevation and local climate (such as the
South Asian monsoon) affecting the mass balance. These factors combine to produce a reverse
ablation gradient, where the point of maximum melt is located several kilometres up-glacier from
the terminus. In times of recession, a low angle, or even reversed, longitudinal surface profile
develops that is hummocky, promoting the surface storage of water in steep-sided supraglacial
ponds. These ponds serve to attenuate flows and regulate the outlet hydrograph. Additionally, in
contrast to their clean-ice counterparts, DCGs convey at least some of their surface water at the
ice-debris interface, likely as a thin film, and the debris layer itself can provide temporary water
storage that delays peak flow at the terminus.
Englacially, channels are likely formed through downcutting and/or the exploitation of
structural weaknesses, and the surface debris layer probably plays an important role in
determining the thermal characteristics of the upper part of the ice column. Subglacially, little is
known, but inferences point to the likely presence of water through the observation of seasonal
velocity speed-ups and bulk meltwater analysis. At the terminus, recent recession has resulted in
the development of moraine-impounded lakes, which are increasing in both number and size in
many areas of the world (Gardelle et al., 2011). Downstream, many millions of people rely on
glacially-sourced water for irrigation, power and sanitation, but a key gap remains in determining
the importance (in terms of quantity and quality) of meltwater as opposed to groundwater and
precipitation with increasing distance from the glacier terminus.
Despite the importance of glacially-sourced meltwater for many populations around the
world, knowledge of the hydrology of DCGs lags behind that of their clean-ice counterparts. In
particular, the subsurface hydrology of DCGs remains largely un-investigated and poorly
understood. Similarly, key parameters governing the formation and structure of these systems,



particularly thermal regime and basal conditions, are also largely unknown at DCGs. On the basis
of the above review, we summarise the current status of our understanding of the hydrology of
DCGs as a schematic illustration in Figure 10.
Inspection of Figure 10 reveals eight candidate areas for future hydrological research,
considered below:
1. **Water flow through and beneath the supraglacial debris layer**. Currently, there has been
minimal research into debris layer hydrology, whether it be a focus on water movement,
water storage, water chemistry, links to other parts of the glacier hydrological system, or
the removal of meltwater from the system through evaporation from the debris layer. Not
only is this important for considering potential delays within the drainage system due to
water storage and the impact upon thermal properties of the debris layer due to the role
moisture holds in dictating thermal conductivity, but it could also have an effect on water
quality downstream. It was noted in Section 5 that water flowing through debris or
sediment, particularly at the bed, entrains greater concentrations of solutes and SSC. With
flowpaths through debris at the surface increasing in extent as both the debris cover
continues to increase upglacier (Kirkbride and Warren, 1999; Stokes et al., 2007) and
meltwater production increases with warming temperatures, these solute levels could be
expected to be raised further, affecting proglacial water quality. Furthermore, the
hydrology of the sub-debris layer ice surface is likely to exert an important influence on
ablation, and thus the production of meltwater.
2. **Supraglacial pond hydrology**. Although substantial recent effort has been directed to the
study of supraglacial ponds and lakes at DCGs, the flow of water within these features, and
between them and other parts of the hydrological system, remains poorly understood, as
does their biogeochemistry (Bhatt et al., 2007; Takeuchi et al., 2012). Meltwater is stored
within supraglacial ponds and lakes - and as more, larger lakes form with greater future
meltwater production - this could delay outflow regimes both diurnally and seasonally.
How water is transported out of a lake is also poorly understood: are there supraglacial or
englacial links between ponds; if they are englacial, is all of this water transported to the
next pond or is some routed deeper into the glacier? As a result, this could influence the
development of englacial and subglacial drainage networks by altering the amount of water
that is, or can be, transported within the glacier.
3. **DCG thermal regime**. An almost complete lack of knowledge of the thermal regime of high-
elevation DCGs has resulted in a critically poor understanding of the existence and
character of englacial and subglacial drainage systems. If water cannot drain into such
glaciers it is unlikely that an englacial system can exist at all. Yet, it is unknown whether
englacial systems are entirely limited by the thermal regime of the ice.
4. **DCG englacial drainage.** Despite detailed glaciospeleological investigations, access has
limited these to large, open channels in accessible areas. Hence, little is known about active
englacial hydrology or smaller englacial hydrological pathways deeper within DCGs, or how
meltwater is transported from the supraglacial system into the glacier. The small scale
(microporous) movement of water between ice crystals has also received very little
attention and may form an important meltwater flowpath, for example through rotted
surface ice. At the larger scale, englacial drainage appears to be governed by base-levels,





but controls over such levels and flow pathway configurations are poorly understood (they
could be local or dictated by proglacial lake level), while englacial drainage below such
levels (i.e. within the phreatic zone) remains un-investigated.
5. **DCG subglacial drainage.** Perhaps the greatest hydrological unknown of DCGs is that of the
existence and character of subglacial drainage, which is critical to governing both ice
motion and meltwater quality. Several indirect studies have suggested the existence of
such effects, but no definitive evidence has yet been reported. If the presence of subglacial
drainage is reported at high-elevation DCGs, then exploring the character and spatio-
temporal variability of such drainage represents a key research priority.
6. **Groundwater flows.** While water loss from DCGs has been inferred, no study has yet
reported on the mechanisms and rates of water transfer between a DCG's drainage system
and that of the underlying substrate. It is therefore important to understand the proportion
of river discharge that is provided by glacier meltwater and runoff, and how much is being
stored within or immediately beyond the glacial drainage network at different time scales.
7. **Long-term water delivery from DCGs**. Long term records of proglacial discharge from DCGs
are scarce, being limited to less than a decade of measurements for a small number of
Himalayan DCGs. As DCGs are predicted to ablate more rapidly with the formation and
growth of more supraglacial ponds and ice cliffs, discharge has been projected to increase
in the short-term but decrease in the long-term, creating concerns for future water
availability in many regions. A greater understanding of how DCGs are and will respond to
the current and future warming climate would constrain future proglacial discharge
volumes and thus help to mitigate water resource issues and other hazards such as
potentially unstable moraine-dammed proglacial lakes.
8. **Local climate influence on DCG hydrology**. Regionally, the local climate is highly likely to
have a substantial influence on the hydrological systems of DCGs, for example, monsoon-
related weather. However, largely due to the inclement weather associated with monsoon
precipitation at high elevations, the hydrological influence of the monsoon has not yet
been addressed. Research to understand the role of monsoon conditions, and its
relationship to non-monsoon conditions, is therefore required.

## 941  8. Author contribution

KM and BH planned the manuscript. KM led the manuscript writing and illustration with all co-
authors contributing to specific sections.

## 944  9. Competing interests

The authors declare that they have no conflict of interest.

## 946  10. Acknowledgements

This research was supported by the 'EverDrill' Natural Environment Research Council Grant
awarded to Aberystwyth University (NE/P002021) and the University of Leeds (NE/P00265X). KM
is funded by an AberDoc PhD Studentship. The authors thank C. Scott Watson (University of Leeds)
for providing the RapidEye image used in Figure 1, and Antony Smith (Aberystwyth University) for



redrawing Figure 9 and illustrating Figure 10. They are also grateful to Himalayan Research Expeditions for organising the logistics that supported fieldwork in Nepal during 2017, and in particular Mahesh Magar for guiding and navigation.

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



## 12. Figures

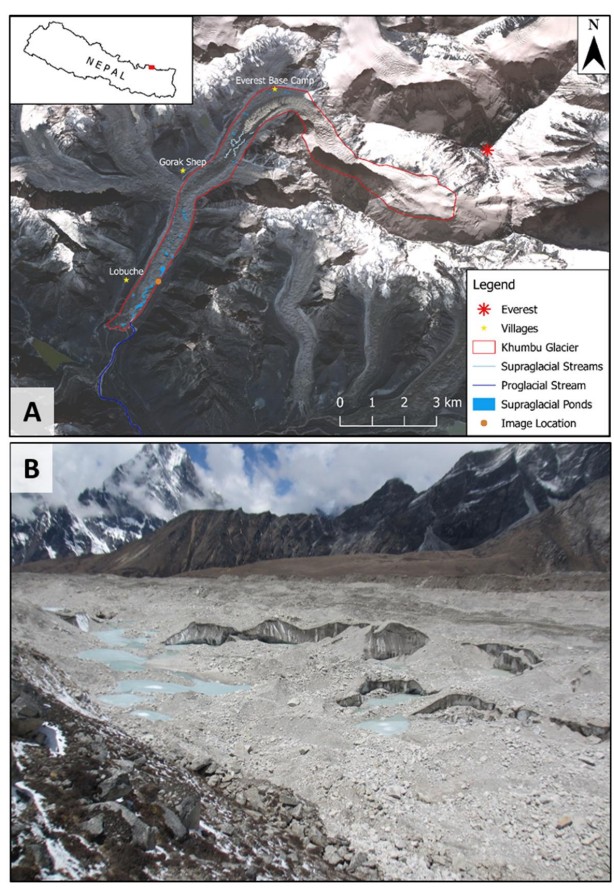


*Figure 1 – An example of a typical and particularly well-studied DCG, Khumbu Glacier, Nepal Himalaya. (A) shows a RapidEye image of the glacier acquired on 17.11.2016 (Planet Team, 2017). The major supraglacial hydrological features (larger supraglacial ponds, supraglacial lakes and any supraglacial streams), the proglacial stream and the location from which the image in (B) was acquired are labelled. (B) shows an oblique photograph looking across the glacier surface (image acquisition location shown in (A), taken in the direction of the glacier terminus), also showing some of the supraglacial ponds as well as ice cliffs and variable surface topography. Image credit: KM*







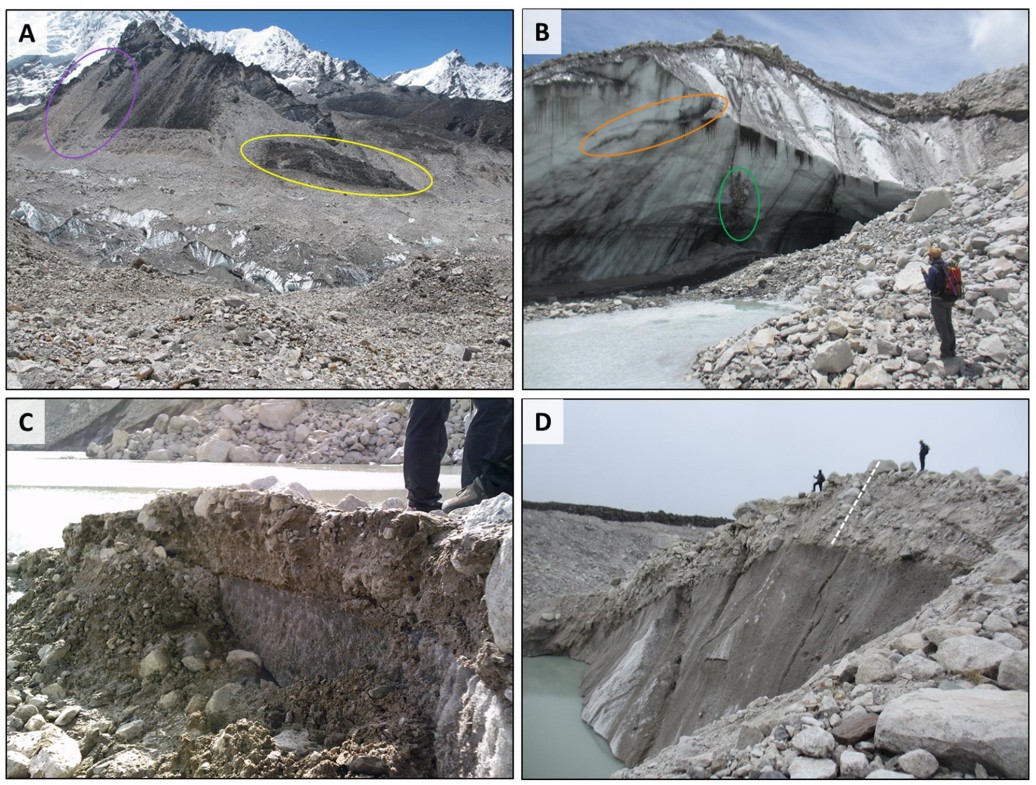

*Figure 2 – Images illustrating variations in debris thickness over Khumbu Glacier, Nepal Himalaya: (A) a landslide scar (yellow circle) and unstable rock faces (purple circle) providing debris to the glacier surface; image is taken looking east across the surface of Khumbu Glacier, and the debris layer above ice cliffs can also be seen. (B) shows an ice cliff with entrained debris (green circle), debris-filled crevasse traces (orange circle), and a moderately-thick debris layer above (~1-2 m); (C) a thin debris layer (~20 cm) above ice adjacent to a supraglacial pond; and (D) a thick debris cover (>5 metres, indicated by the white dashed line) above an ice cliff. Image credits: (A) DQ and (B-D) KM*






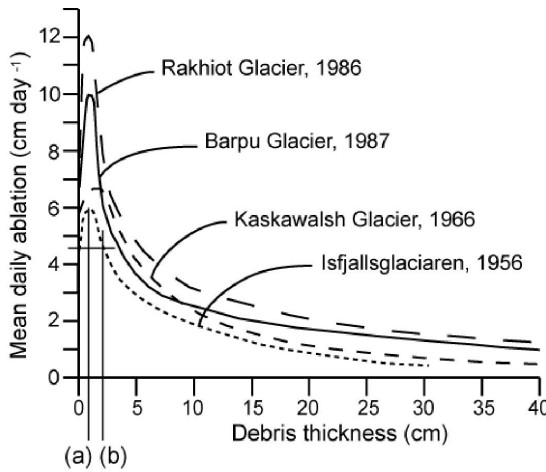


*Figure 3 - Østrem curve examples from Nicholson & Benn (2006, and citations therein), showing the variations in the relationship between debris thickness and ice ablation on different glaciers*


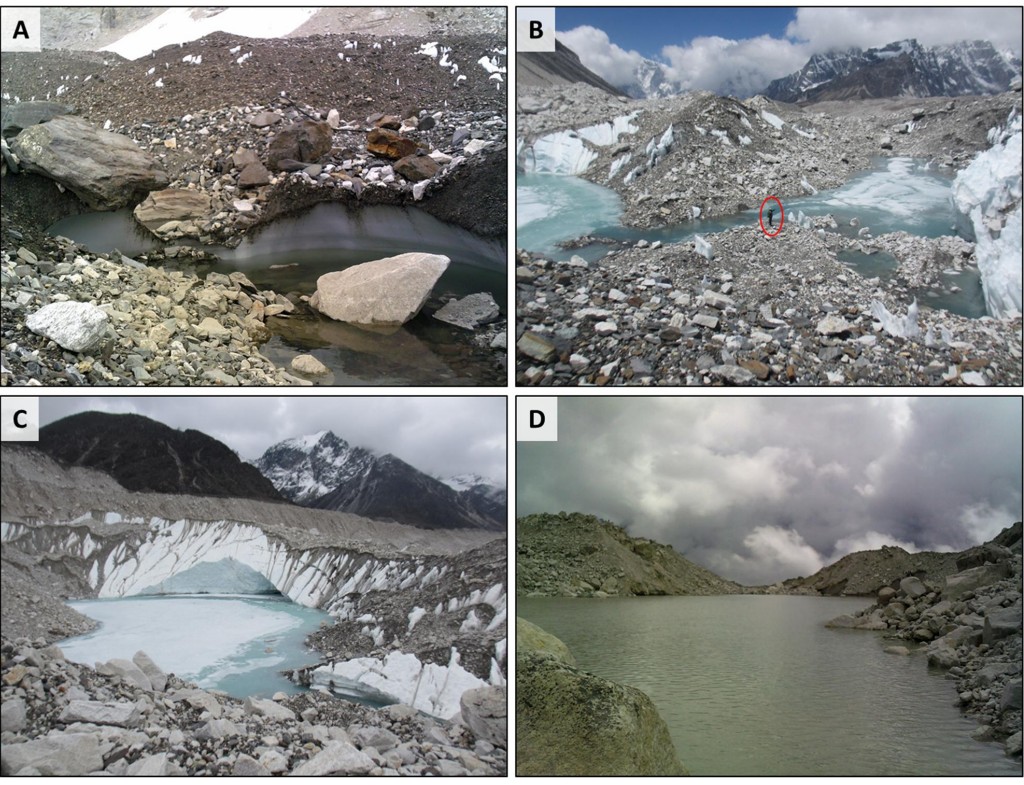



*Figure 4 – Examples of supraglacial ponds on Khumbu Glacier, Nepal Himalaya, ranging in diameter from several metres (A), to tens of metres (B, C) and hundreds of metres (D). (A) and (C) also feature a notably large adjacent ice cliff system, relative to each pond/lake size; while (B) has a cliff system on the far of each of the lake sides (a person is circled for scale). Image credits: KM*


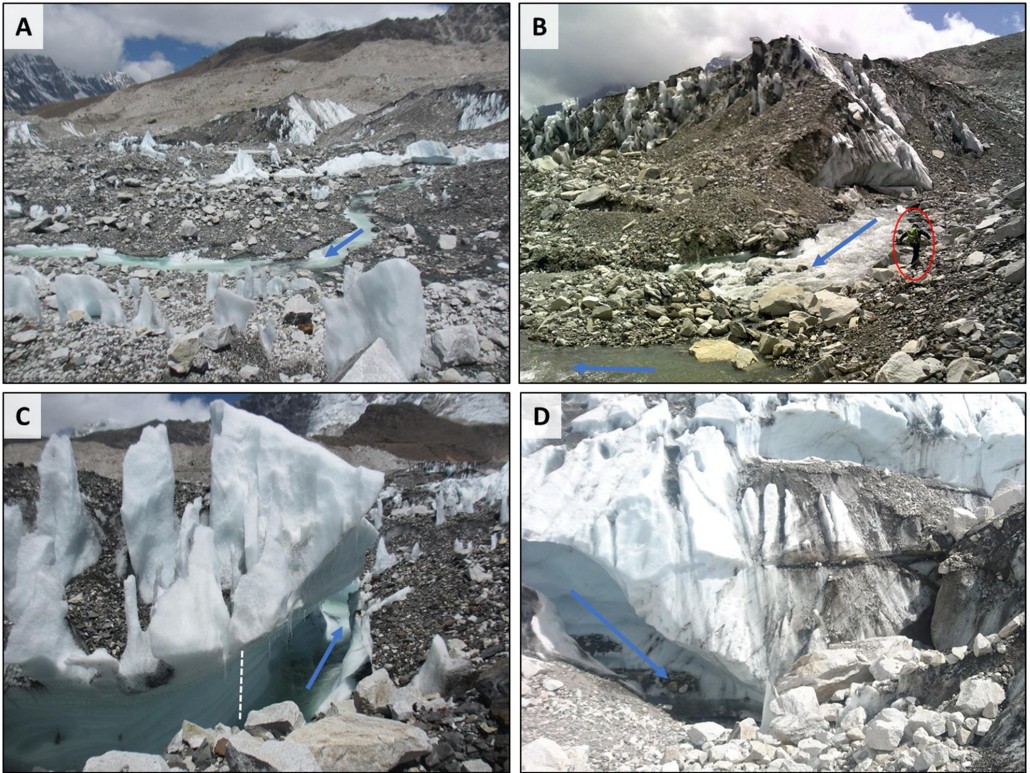


*Figure 5 - Examples of a large supraglacial stream on Khumbu Glacier, Nepal Himalaya; blue arrows indicate water flow direction. (A) shows the stream in the upper ablation area; (B) shows the stream again, approximately 2 km downstream of (A) in the central ablation area and nearly twice the volume, just above a confluence with another large stream (bottom left of image; person shown for scale); (C) is an example of multiple levels of downcutting of the stream (grooves indicated by white dashed line, ~1 m in height), slightly upglacier in location from (A); and (D) shows where this stream eventually disappears below the surface to become englacial, after several hundred metres of progressive downcutting, visible from the multiple relict levels. The drop in the channel is ~10 m, with the stream dropping another few metres beyond the boulders to the right of the image. Image credits: (A-C) KM; (D) EM*




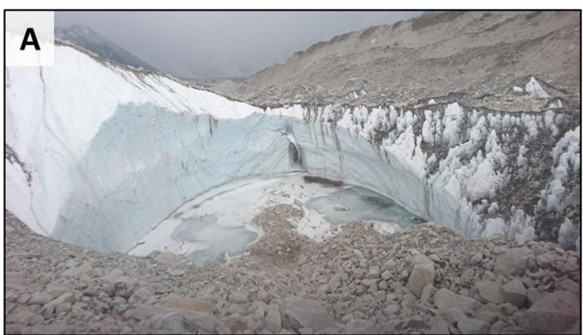
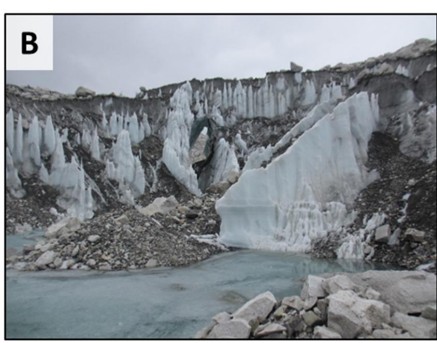


*Figure 6 – A relict englacial feature in the centre of an ice cliff on Khumbu Glacier, Nepal Himalaya, viewed (A) from upglacier, and (B) from downglacier. The associated supraglacial pond is hypothesised to have drained through this feature in the past. Following the drainage event, the pond water-level would have dropped, exposing the ice cliffs around its edge and resulting in the pond water-level being too low to sustain a water flow through the channel. On the downglacier side (B) a vast amount of surface lowering has occurred and the previously englacial channel is now visible from the surface. The relict channel could be seen to continue to meander and downcut for around 200 m further downglacier until joining a pond. The englacial feature is approximately 10 m in height. Image credits: (A) EM; (B) KM*


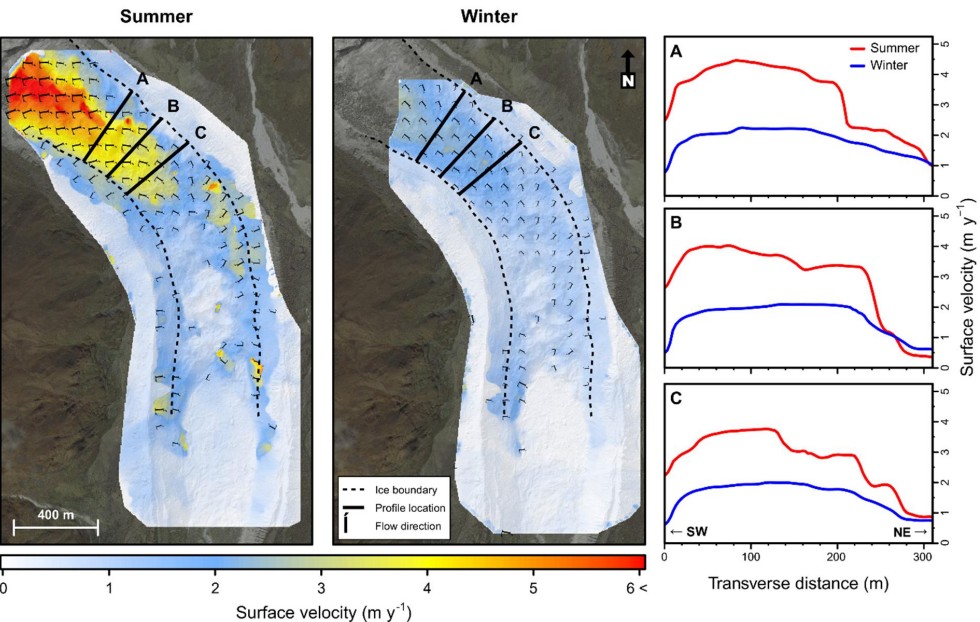


*Figure 7 – Surface velocity maps of Lirung Glacier, Nepal Himalaya, from Kraaijenbrink et al. (2016b) during summer (left) and winter (right), with three transverse velocity profiles (A-C) at the locations marked. Available under a Creative Commons Attribution 4.0 License*





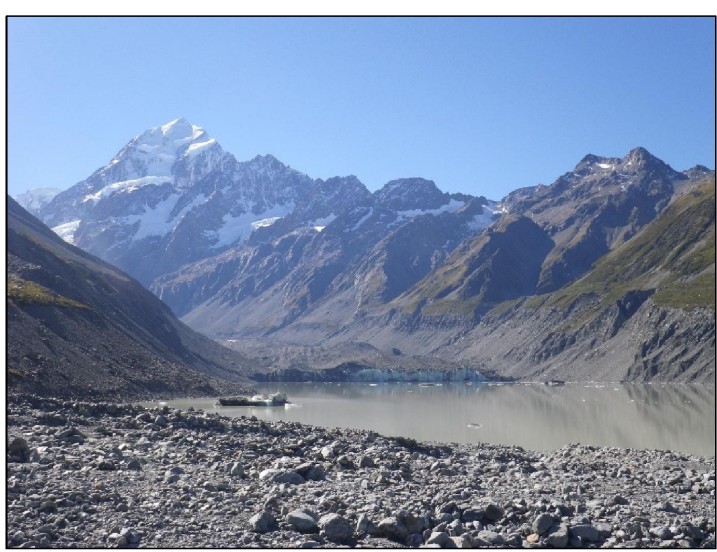


*Figure 8 – Image of Hooker Lake, a proglacial lake in front of the debris-covered Hooker Glacier, New Zealand Southern Alps, taken in 2013. For scale, the ice cliff at the terminus of the glacier is ~30 m in height. Image credit: TIF*






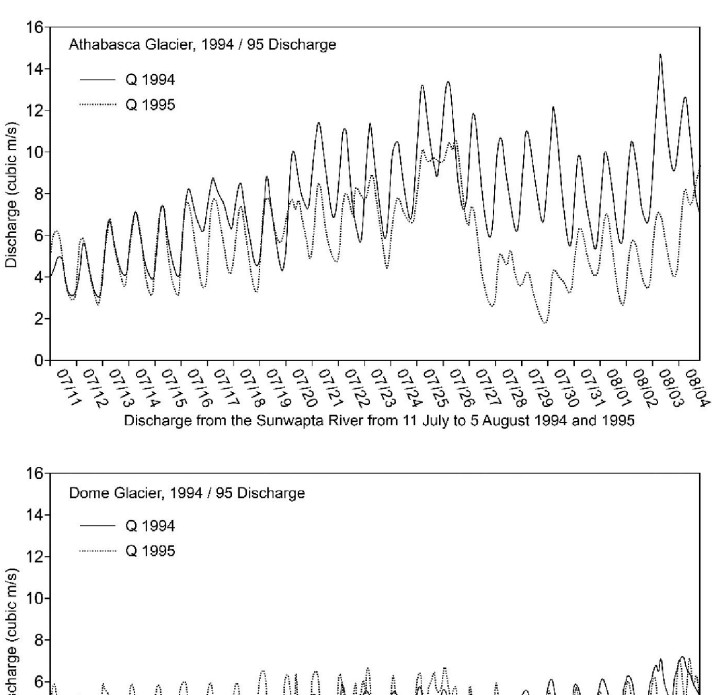


*Figure 9 – Hydrographs of proglacial discharge of the clean-ice Athabasca Glacier and the adjacent debris-covered Dome Glacier, Canadian Rockies, over the ablation months of July and August 1994 and 1995. Figure redrawn from Mattson (2000)*







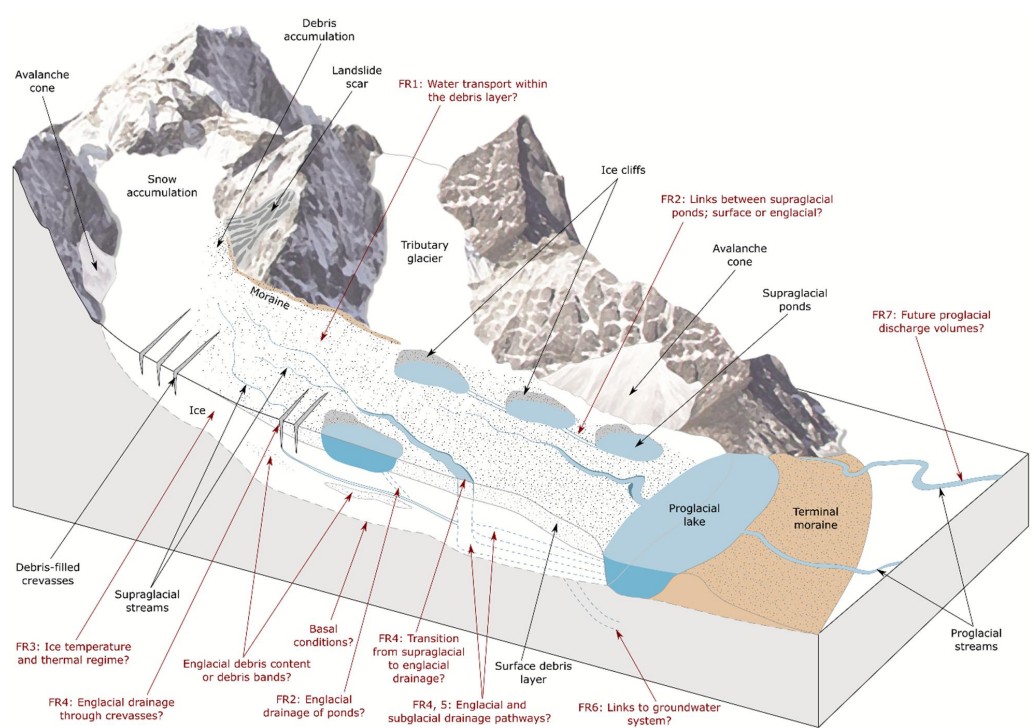

*Figure 10 – A conceptual illustration of the hydrological system of a DCG, including all known (black text), poorly understood and completely unknown potential hydrological features, highlighted in red text and linked to the Future Research (FR) areas for future hydrological research*
