# Peer review of "Review article: The hydrology of debris-covered glaciers - state of the science and future research directions"

_The Cryosphere, 2017_

## Short Comment (SC1) · 30 Nov 2017

I found very interesting the tentative of Miles at al. to review the "hydrology of debris glaciers". The topic is large, probably too much.

Therefore, I hope that many experts will contribute to this discussion.

Personally, I am not expert in everything written here, so I am providing below just my contribution in relation to what I wrote in the recent years.

**Suggestions:**

Section 2:

Concepts like "The areal extent of glacial debris cover in the Himalayan region is increasing and predicted to expand further (Bolch et al., 2008; Rowan et al., 2015; Thakuri et al., 2014)" should be added in this section.

Line 112: Salerno et al., 2017 found that debris thickness is higher for gentle downstream surface (ablation zone) gradients supposing that this condition favors the accumulation of debris (less gravitational stresses).

Line 149 to 151:

I think that the follow evidences should be treated/included/or discussed:

Recent large-scale geodetic studies based on remotely sensed data have provided evidence that the present-day surface lowering rates of some debris-covered glacier areas in the Hindu-Kush–Himalaya may be similar to those of debris-free areas even within the same altitudinal range (e.g., Kääb et al., 2012; Nuimura et al., 2012; Gardelle et al., 2013; Pellicciotti et al., 2015; Ragettli et al., 2016; Salerno et al., 2017).

Therefore, I think that the effect of debris on the surface mass balance of glaciers remains unclear.

*Line 181: I suggest:*

Salerno et al., 2017, analyzing a wide population of glaciers with a stochastic approach, found that the debris coverage and thickness are not significantly responsible for the development of supraglacial ponds, the elevation changes, or the shift in SLAs. At this regards they observed that the main morphological factor controlling the debris glacier water balance under stressed climatic conditions is the glacier surface gradient and in particular the surface gradient of the downstream portion of the glacier. From a physical point of view, lower surface gradients are thought to induce reduced glacier ice flow, thus allowing the development of stagnant ice (e.g., Scherler et al., 2011). Under these conditions, consequent lower terminus retreat rates have already been observed (e.g., Bolch et al., 2008; Scherler et al., 2011), as well as the development of supraglacial ponds (e.g., Reynolds, 2000; Quincey et al., 2007; Sakai, 2012; Salerno et al., 2012. In this analysis, Salerno et al., 2017 note that downstream surface gradients over 15° inhibit glacier surface lowering, while the greatest surface lowering is found on downstream surface gradients lower than 5°.

*Racoviteanu, A.E., Arnaud, Y., Williams, M.W., Manley, W.F., 2015. Spatial pat-terns in glacier characteristics and area changes from 1962 to 2006 in the Kanchenjunga–Sikkim area, eastern Himalaya. Cryosphere9, 505–523. http://dx.doi.org/10.5194/tc-9-505-2015.*

*Salerno, F., Thakuri, S., Tartari, G., Nuimura, T., Sunako, S., Sakai, A., & Fujita, K. (2017). Debris-covered glacier anomaly? Morphological factors controlling changes in the mass balance, surface area, terminus position, and snow line altitude of Himalayan glaciers. Earth and Planetary Science Letters, 471, 19-31.*

*Loibl, D.M., Lehmkuhl, F., Grießinger, J., 2014. Reconstructing glacier retreat since the Little Ice Age in SE Tibet by glacier mapping and equilibrium line altitude calculation. Geomorphology214, 22–39. http://dx.doi.org/10.1016/j.geomorph.2014.03.018.*

Line 187-191:

Salerno et al., 2017 show that where supraglacial ponds develop, the glaciers register further surface lowering.

However, other authors consider that the insulating effect of debris cover has a larger effect on total mass loss than the enhanced ice ablation due to supraglacial ponds and exposed ice cliffs (e.g., Hambrey et al., 2008;Vincent et al., 2016).

Vincent, C., Wagnon, P., Shea, J.M., Immerzeel, W.W., Kraaijenbrink, P., Shrestha, D., Soruco, A., Arnaud, Y., Brun, F., Berthier, E., Sherpa, S.F., 2016. Reduced melt on debris-covered glaciers: investigations from Changri Nup Glacier, Nepal. Cryosphere10, 1845–1858. http://dx.doi.org/10.5194/tc-10-1845-2016.

Hambrey, M.J., Quincey, D.J., Glasser, N.F., Reynolds, J.M., Richardson, S.J., Clemmens, S., 2008. Sedimentological, geomorphological, and dynamic context of debris-mantled glaciers, Mount Everest (Sagarmatha) region, Nepal. Quat. Sci. Rev.27, 2361–2389. http://dx.doi.org/10.1016/j.quascirev.2008.08.010.

Personally, I believe (and wrote) about the effects of supraglacial ponds on melt, but other authors less. Therefore I think that, generally, concepts like "Supraglacial ponds are responsible for a large proportion of the melt from DCGs" (e.g., line 382, but anywhere in the paper) should be states as e.g., "many authors currently think......".

Line 795 Water quality needs probably a separate paragraph.

I think that In this contest from an ecological prospective it is important to reference here the recent paper of Salerno et al., 2016 published on unique data set of twenty years of chemical data coupled on field meteorological data at high elevation in Himalaya. They found that debris glacier retreat likely is the main factor responsible for the observed increase of solute concentrations of proglacial surface waters in the last two decades. The temperature of April is the effective drive of the observed enhanced glacier melting process, the main factor responsible for the observed increase of sulfate concentrations. These chemical variations (sulfate at a higher extent) represent a response of these fragile ecosystems to climate change. Even if these changes do not pose a direct and immediate threat to the biota, they occurred in a limited time span, and significantly modified the average chemical composition of lake water. For these reasons, the lakes and the main factors driving their variability should be regularly monitored in the future, also in relation to the lake role as ecosystem services.

Salerno, F., Rogora, M., Balestrini, R., Lami, A., Tartari, G. A., Thakuri, S., ... & Tartari, G. (2016). Glacier melting increases the solute concentrations of Himalayan glacial lakes. Environmental science & technology, 50(17), 9150-9160.

**Minor suggestions:**

I suggest to reference the recent work of Tristram et al., 2018. (Tristram et al., 2018. Supraglacial ponds regulate runoff from Himalayan debris-covered glaciers. JGR).

Line 276 and 282: the right reference is Salerno et al., 2017 not Salerno et al., 2015.

Line 797 Please reference as model applied o single glaciers even Soncini et al., 2016.

---

## Referee Comment (RC1) · Anonymous Referee #1 · 11 Dec 2017

**Review of "Review article: The hydrology of debris-covered glaciers – state of the science and future research directions" by K.E. Miles et al.**

General

In this manuscript, the authors aim to give a review of the state of knowledge of the hydrology of debris-covered glaciers and suggest some directions for future research. Debris-covered glaciers are an important topic of current research, since there are many open questions surrounding the response of such glaciers to changes in climatic conditions. A better understanding of hydrological processes is of particular significance due to the importance of downstream water supply from glaciated regions that contain a large proportion of debris-covered glaciers.

While I personally found many aspects of this article interesting, unfortunately I do not recommend it to be accepted for The Cryosphere in its current form, for the following reasons:

1)      Good review papers are more than surveys of previous findings. They provide original insights and analysis that help to crystallize the most important concepts and current state of a field. However, this article reads more like a literature review than a review paper. There is no attempt to frame the discussion by presenting the key physical processes at the beginning of each section, nor is there a clear synthesis of the information at the end of each section. The choice of figures is also not so helpful, as many are photographs taken by the authors, which although original, do not help to elucidate the processes being discussed the way that clear schematic drawings would. These shortcomings limit the utility of the manuscript.

2)      The manuscript does not clearly highlight the conditions under which hydrological features are expected to be different for debris-covered glaciers. In many places, the results of previous studies on debris-covered glaciers are presented without explicitly discussing whether they are consistent with similar results from studies of clean ice glaciers. This leaves a confusing impression, especially since one important goal of this review paper should be to link well-known results from previous studies of clean ice glaciers to the relatively new results for debris-covered glaciers.

3)      The section on candidate areas for future research needs considerable improvement. Several of the items on the list are either not well connected to the rest of the article or else difficult to motivate as research priorities. Also, there is no discussion of how the research might be carried out, so the list appears more aspirational than practical.

Example problems

Pages 7-10 (Section 3.3 – Supraglacial ponds)

This section has a lot of good material on an interesting topic but it is poorly organized. No overall discussion of the physical processes is presented but rather bits and pieces that come up along the way. This leads to some confusion in the order of the presentation, as some controls on ponding (lines 265-280) are discussed before the main mechanisms of pond formation are outlined, which seems backwards. There is no

synthesis of the main ideas, nor are any open questions raised at the end of the section. The one figure chosen (Figure 4 on p.43 with caption on p.44) does not help the reader to better understand why/how supraglacial ponds form, grow, or drain.

As an example of a better structure for this and other sections, the authors might wish to consider one of the clean-ice hydrology reviews that they cite as a previous comparable study (p.4, lines 129-130), the clearest being the classic paper by Fountain and Walder (1998).

Pages 11-14 (Section 4 – Englacial hydrology)

An argument is made (line 436, p.11) that "a glacier's thermal structure determines the water content of englacial ice…" but no studies presented have shown any difference in the thermal structure of debris-covered glaciers compared to nearby clean ice glaciers. Hence, it does not make sense to focus much on this and certainly not as the first main topic of the section.

There is substantial overlap between debris-covered and clean ice glaciers for this section but without a review of the state of clean ice englacial hydrological knowledge, it is challenging to understand what is important and why. There is only limited utility in a detailed discussion of debris-covered glacier observations that are consistent with those observed on clean ice glaciers and these can just be mentioned in passing.

The three formation mechanisms for englacial channels within DCGs (lines 509-542) would be better placed at the beginning of the section, as part of an overview of the key processes. A schematic figure of these mechanisms would go a long way to making this section clearer.

Pages 14-16 (Section 5 – Subglacial hydrology)

There is a very cursory and incomplete discussion of what might be different about the basal hydrology of debris-covered glaciers as compared to clean ice glaciers in lines 576-585 and there is no description of the key processes of subglacial drainage theory here. All of the studies on debris-covered glaciers summarized in the rest of the section (lines 586-638) could be equally true for clean ice glaciers. Although there are no conclusive studies to report, it would be still useful to discuss how differences in temporal evolution of surface and englacial hydrological processes, water storage located throughout the glacier, and the increased likelihood of finding basal sediment might alter expectations for the subglacial drainage of debris-covered glaciers.

Pages 22/23 (Candidate areas for future hydrological research)

Some of the items on this list are not clearly defined and in general, there is no explanation of how this research might be carried out, not even a brief sketch of how one might start to set up future work. It would be good to identify, for example, where existing data or models are weak or incomplete, how this inhibits current understanding, and what explicit steps future researchers should do to address this. For example: what to measure and how; what future theoretical or numerical models should include; and which papers might serve as a starting point for the work.

In addition, some other specific problems are:

Item 4: The focus on small-scale movement of water between ice crystals comes out of nowhere. It does not appear anywhere in the rest of the article and there is no citation, so it is unclear why this might matter or what previous studies have led to this idea.

Item 5: It is still not so clear why one should expect anything different for the subglacial drainage of debris-covered glaciers or how the authors envisage future work that could shed light on this matter.

Item 7: Long-term water delivery from debris-covered glaciers is intimately connected to mass balance, ice dynamics, and the response to climatic changes. It does not make sense to discuss this topic purely in a hydrological context. It is also unclear if future work is meant to be modelled numerically or studied observationally or both.

Item 8: "Research to understand the role of monsoon conditions, and its relationship to non-monsoon conditions, is therefore required" is confusing as this appears to be less about understanding debris-covered glaciers and more about understanding the effects of one particular climate's local influences on all glaciers. Generalizing to local climate influences on global debris-covered glaciers is similarly unclear.

---

## Referee Comment (RC2) · Anonymous Referee #2 · 2 Jan 2018

This paper provides a summary of the current state of knowledge about the hydrological systems of debris covered glaciers. The paper is well written and researched, and I found it quite interesting to read. However, I am not convinced of its merits as a review paper in The Cryosphere. The paper is more of a literature review, such as might be found in a thesis or a grant proposal (as such, it is good), rather than a 'review article' in which I would expect to see more synthesis based on the literature (rather than a summary of what is in the literature).

A second issue is that I am not convinced the field of debris-covered-glacier hydology is sufficiently well developed (as distinct from glacier hydrology more generally)

[Figure]

to warrant a review paper. This manuscript makes frequent comment about how little is known, especially the sections on englacial and subglacial hydrology, and that leads to considerable speculation. If so little is known, why a review paper? In places the review focusses more broadly on debris-covered glaciers, and at times more on 'Himalyan glacier hydrology' (eg, the introduction is mostly taken up with disucssion of Asian water security and how debris-covered glaciers are expected to evolve with climate change).

I can understand that the authors want to demonstrate a need for more work to study such glaciers, but I think there needs to be a clear dicussion of why the hydrology (in particular) of these glaciers is important. In particular, why and in what way, are the en- and sub-glacial systems of a debris-covered glacier thought to be different from other glaciers. It seems clear that the supra-glacial hydrological system is quite different, but a lot of the disucssion of this is tied up with the surface energy and mass balance - and would seem to fit better within a review of the mass balance of debris-covered glaciers rather than the hydrology per se.

I would suggest that the paper might be better framed with a focus just on supraglacial and near-surface hydrology of debris-covered glaciers. This would allow for an expanded and in-depth discussion of this area, on which there has been more work and for which it is clear that the debris cover is important. This could include the effects of the debris cover on the proglacial discharge hydrograph, but I think this needs to have greater emphasis on what is different from other glaciers and why a different treatment is needed.

Specific comments

Section 2, paragraph beginning on l129 - some of the differences from clean-ice glaciers are discussed here, but apart from the supraglacial and near-surface englacial drainage this seems largely speculative or is not specific to the debris-cover (possibile presence of cold ice, low hydraulic potential gradients, presence of proglacial lake,

monsoon-dominated climate).

Section 3.2 - much of the discussion here is presented rather speculatively - 'a similar situation may hold...', 'these situations could be plausible within the debris layer...', 'meltwater could augment the melt of glacier ice...'. These seem quite obvious comparisons to permafrost and proglacial environments with underlain ice, but could it be made more definite what the similarities and/or differences might be for supraglacial debris. Eg. what is the permeability of the debris layer, does the continual release of debris melting out of the ice make a difference?

Section 3.4 - it would be helpful to have more detailed comparison with clean-ice glaciers here. How much less common are supraglacial streams on debris covered ice than clean ice? In what way are they different? (the discussion the final paragraph seems inconclusive as both more and less crevassing are implied on different regions of the glacier).

Section 4 - the statement on l436 is debatable. The thermal structure may certainly influence the formation of an englacial hydrological system, but it is not clear that it 'determines the water content', which suggests a direct relationship. Water is commonly transferred englacially through cold ice in Greenland and Arctic glaciers. It is therefore not clear why knowing the thermal structure of debris-covered glaciers is so important here. It would again help to explain why the thermal structure of these glaciers is expected to be different from other glaciers (does the presence of debris on the surface have a greater effect than other factors such as alitude, accumulation rate, etc?).

Section 5 - It would help to be explicit about why the subglacial drainage systems of debris-covered glaciers are different to other glaciers, or at least to discuss based on some physcial arguments why they might be different. Most of the studies quoted don't sound dissimiliar from what might be found studying non debris-covered glaciers.

l581 - why does a proglacial lake increase the likelihood of some form of subglacial drainage?

Section 6, l660 - is this section suggesting that debris-covered glaciers necessarily go through a cyclic behaviour involving growth and recession, purely through their internal dynamics, or is it referring particularly to glaciers that are retreating for climatic reasons? In regime two, 'surface lowering has begun', suggesting the glacier is not in balance, but it's not clear if this is thought to be \*due\* to the formation of the proglacial lake, or due to other external factors.

l747 - should this say 'variations in discharge' are muted? I couldn't see why discharge itself should be muted (other than due to differences in surface mass balance). The figure looks like it has a variance larger than the 1% quoted in the text - is this correct? Is it clear that the differences between these glaciers are due to the debris cover and not due to other factors (differences in catchment areas and travel times etc)?

Section 7 - the first paragraph here would have been good in the introduction.

l900 - this seems to be divorced from the earlier discussion and references. From what was described earlier it seems clear that there \*is\* some englacial and subglacial drainage?

For points 4,5 and 6, it would be helpful to hypothesise how these are likely to be different from other glaciers, on which more work has been carried out.

Figures - most of the figures are field photos. I think it would help to replace some of these with more schematics that demonstrate the \*processes\* discussed (more specific details than in figure 10).

---

## Author Comment (AC1) · 15 Feb 2018

We thank both reviewers and, given the broad nature and overlap of the suggestions and limitations raised, we address both together.

Summary of Reviews

Specific comments aside (all of which we are willing and able to address), both reviewers express concerns with the structure and content of the manuscript in its present form and ask directly or indirectly for the following.

1. Greater clarity in terms of how the hydrology of debris covered glaciers may be

expected to differ from that of 'clean ice' glaciers, to include greater representation of the latter.

2. The review to extend beyond a literature summary, and for it to provide added value to the reader.

3. The final section on research priorities to follow more directly from the review (and to include an indication of the methods that might be most appropriate for addressing these unresolved issues [Rev #1]).

Authors' Response

We believe strongly that there is substantial merit in a review of the hydrology of high-elevation debris covered glaciers (DCGs). First, we are convinced that the hydrology of high-elevation DCGs is sufficiently distinctive (expanded upon below) to warrant a dedicated review. It is notable that none of the existing widely recognised reviews of glacier hydrology explicitly addresses the hydrology of DCGs. Second, with some notable exceptions, the community of researchers working on high-elevation DCGs tends not to overlap with that researching clean-ice glacier hydrology. We believe a review of DCG hydrology – hitherto not provided – would be of use to both communities and hopefully lead to greater research integration. Part of the problem in providing such a review is that relatively little is known about the hydrology of high-elevation DCGs: certain hydrological processes operating at clean-ice glaciers presumably hold at DCGs (such as englacial water ingress via crevasses and micro-scale inter-crystalline permeability), but others are unknown (such as the details of water flow beneath a thick surface debris layer, the formation and effectiveness of moulins, the role of surface ponds in terms of delivering meltwater to the glacier base, and even whether the glacier base is temperate and therefore hosts an effective drainage system at all). Therefore, the nature of the problem is such that a review of high-elevation DCG hydrology will largely be pointing to what is currently unknown or poorly-known relative to our understanding of lower-elevation clean-ice glacier hydrology. However, we accept that, despite this

requirement for a review of the hydrology of DCGs, our initial manuscript could have provided this better. We agree with the reviewers in this regard and propose below several substantial revisions to the manuscript that we believe will result in it providing a valuable review.

1. The revised manuscript will focus explicitly on Himalayan debris-covered glaciers (changing the title accordingly), allowing (i) clearer separation between the hydrology of the DCGs reviewed and that of contrasting clean-ice glaciers, and (ii) the special hydrological influence of the monsoon as well as that of the low- or reversed-angle glacier tongue (and its consequences) to be considered explicitly. The revised manuscript will refer to debris covered glaciers outside the Himalayas only where directly relevant.

2. The Introduction will now include a section dedicated to how debris-covered glaciers differ fundamentally from clean ice glaciers. A number of key influences will be identified, including: ice being sourced from extremely high elevations (and therefore assumed to be cold); a steep surface gradient, usually including an ice-fall, in the upper ablation area; a very low or reversed gradient debris-covered tongue in the lower ablation area; an inverted mass balance regime across the ablation area as a whole; the presence of linked supraglacial lakes in the lower ablation area; the frequent presence of a proglacial lake; the common presence of a substantial thickness of subglacial morainic deposits; and the hydrological role of the monsoon. All of these will be expected to influence hydrology in some way, but most remain to be evaluated.

3. The subsequent four main sections - addressing supraglacial, englacial, subglacial, and proglacial hydrology - will each be restructured into three sub-sections: (i) a brief summary of what is known on the basis of research at clean-ice glaciers, followed by (ii) a thorough summary of what is known at Himalayan DCGs, and finally (iii) an evaluation of what is not known (but needs to be) at Himalayan DCGs, identified with respect to the distinctive characteristics of DCG hydrology noted in the Introduction. This restructuring into formal sub-sections will add clarity to the review and lead more logically into the subsequent discussion of avenues for future research. Thus, for example, all

features identified as priorities on Figure 10 will now have been considered explicitly earlier in the manuscript.

4. The revised manuscript will include brief suggestions relating to how each of the avenues identified for future research might be carried out.

5. We will replace some of the more specific photographic figures with schematic illustrations.

We believe that, taken together, these revisions will provide a clear and valuable review that also provides substantially more added value than a straightforward summary of the literature - as requested by the reviewers.